# Root water uptake patterns are controlled by tree species interactions and soil water variability

Gökben Demir[1], Andrew J. Guswa[2], Janett Filipzik[1], Johanna Clara Metzger[1,3], Christine Römermann[4,6], Anke Hildebrandt[1,5,6]

[1] Group of Terrestrial Ecohydrology, Institute of Geoscience, Friedrich Schiller University Jena, Jena, 07749, Germany
[2] Picker Engineering Program, Smith College, Northampton, MA, 01063, USA
[3]Institute of Soil Science, University of Hamburg, Hamburg, 20146, Germany
[4]Plant Biodiversity, Institute of Ecology and Evolution, Friedrich Schiller University Jena, Jena, 07743, Germany
[5] Department of Computational Hydrosystems, Helmholtz Centre for Environmental Research – UFZ, Leipzig, 04318, Germany
[6]German Centre for Integrative Biodiversity Research (iDiv) Halle -Jena-Leipzig, Leipzig, 04103, Germany

Correspondence to: goekben.demir@uni-jena.de and anke.hildebrandt@ufz.de

## Abstract

Throughfall is the largest source of water entering the soil in forests, and its spatial distribution depends on several biotic and abiotic factors. It is well documented that the distribution of throughfall results in reoccurring higher and lower water inputs at certain locations. However, the role of horizontal root water uptake patterns in understanding the effects of throughfall patterns on subsurface water dynamics remains unresolved. Therefore, here we investigate root water uptake patterns by considering spatial patterns of throughfall and soil water patterns in addition to soil and neighboring tree characteristics. In a beech-dominated mixed deciduous forest in a temperate climate, we conducted weekly intensive throughfall sampling at locations paired with soil moisture sensors during the 2019 growing season. We employed a linear mixed effects model to understand controlling factors for root water uptake patterns. Our results show that soil water patterns and interactions among neighbouring trees are the most significant factors regulating root water uptake patterns. Temporally stable throughfall patterns did not influence root water uptake patterns. Similarly, soil properties were unimportant for spatial patterns of root water uptake. We

found that wetter locations (rarely associated with throughfall hotspots) promoted greater root water uptake. Root water uptake in monitored soil layers also increased with neighbourhood species richness. Ultimately our findings suggest that complementarity mechanisms within the forest stand, in addition to soil water variability and availability, govern root water uptake patterns.

**Key words**: root water uptake, throughfall, soil water, spatial patterns, beech

# 1) Introduction

Vegetation intercepts and redirects precipitation into throughfall and stemflow, collectively referred to as below-canopy precipitation. Moreover, throughfall is usually the largest component of below canopy precipitation (Levia and Frost, 2006; Sadeghi et al., 2020). For instance, in temperate forests throughfall can account for about 70% of above canopy precipitation (Levia and Frost, 2003; Sadeghi et al., 2020). This makes it the primary source of soil moisture replenishment in vegetated areas.

Below-canopy precipitation is modified by several biotic and abiotic factors (Levia and Frost, 2006; Levia et al., 2011), such as vegetation type and canopy architecture (Crockford and Richardson, 2000; Pypker et al., 2011; Levia et al., 2017), forest structure (Rodrigues et al., 2022), meteorological elements such as wind speed (Staelens et al., 2008; Van Stan et al., 2011; Fan et al., 2015), precipitation intensity and event size (Dunkerley, 2014; Magliano et al., 2019; Zhang et al., 2016; Staelens et al., 2008). This implies that it inherently varies across space and time. Furthermore, previous studies showed that the spatial distribution of throughfall persists repeatedly over time (Keim et al., 2005; Staelens et al., 2006; Guswa and Spence, 2012; Carlyle-Moses et al., 2014; Metzger et al., 2017; Van Stan et al., 2020).

Throughfall patterns potentially translate their spatial variability into soil moisture (Raat et al., 2002; Blume et al., 2009; Zimmermann et al., 2009; Zehe et al., 2010; Bachmair et al., 2012; Rosenbaum et al., 2012; Zhang et al., 2016). A decade ago Coenders-Gerrits et al., (2013) proposed that throughfall patterns are translated into soil wetting dynamics with a model based on combined hillslope topographic and throughfall data collected in a beech-dominated catchment. However, in this model, the effect of throughfall patterns on soil moisture patterns rapidly ceased. Later, Metzger et al. (2017) empirically

confirmed that throughfall patterns barely alter soil moisture in response to rainfall and the limited influence rapidly disappears. Recently, Zhu et al. (2021) observed that stable spatial patterns of throughfall were related to the spatial distribution of soil moisture. However, this relationship was restricted only to relatively wet soil locations and throughfall hotspots. They also showed that throughfall patterns had a weak influence on the temporal dynamics of soil water content compared to soil bulk density and litter layer properties.

Previously proposed explanations for the weak and short-term influence of throughfall patterns on the soil moisture patterns include: soil properties (Metzger et al., 2017), preferential flow induced by dry antecedent soil conditions (Jost et al., 2004; Blume et al., 2009; Molina et al., 2019; Fischer et al., 2023), litter layer (Raat et al., 2002), and local root water uptake enhanced by throughfall hotspots (Bouten et al., 1992; Coenders-Gerrits et al., 2013). Based on a one-dimensional soil water model, Bouten et al. (1992) proposed that throughfall patterns alter and localize root water uptake and promote fast drainage via preferential flow paths. However, to the best of our knowledge, the feedback mechanism of throughfall patterns on root water uptake variation has not yet been investigated in the field. Therefore, it is unclear how water uptake patterns play a role in translating throughfall patterns into spatio-temporal variation of soil water and vice versa.

Soil water distribution may shape root water uptake patterns even more than root networks (Kühnhammer et al., 2020). Soil properties control soil water redistribution (Grayson et al., 1997; Cosh et al., 2008; Jarecke et al., 2021) and water availability (Vereecken et al., 2007; Cai et al., 2018).Thus soil properties can influence root water uptake patterns (Nadezhdina et al., 2007; Kirchen et al., 2017). Moreover, variations in soil water content reflect water uptake by root systems (Hupet et al., 2002; Schume et al., 2004; Schwärzel et al., 2009; Guderle and Hildebrandt, 2015; Jackisch et al., 2020). On the flip side, root water uptake can amplify but mostly homogenize soil moisture distribution (Hupet and Vanclooster, 2005; Teuling and Troch, 2005; Ivanov et al., 2010; Baroni et al., 2013; Martínez García et al., 2014). Root networks can also regulate soil moisture distribution by transporting water from wetter places to drier locations, which has been observed in a variety of ecosystems (e.g., Emerman and Dawson, 1996; Katul and Siqueira, 2010; Yu and D'Odorico, 2015; Priyadarshini et al., 2016; Hafner et al., 2017).

In addition, tree species richness affects the dynamics of root water uptake (e.g., Volkmann et al., 2016; Spanner et al., 2022). Neighboring different tree species utilize different hydraulic strategies, such as extracting water from different soil depths (Silvertown et al., 2015; Guo et al., 2018; Brum et al., 2019). However, soil water scarcity can initiate or enhance competition mechanisms for water among tree species (González de Andrés et al., 2018; Vitali et al., 2018; Magh et al., 2020). Moreover, studies conducted in temperate forest ecosystems demonstrate that the relationship between tree species richness and water uptake mechanisms varies (Krämer and Hölscher, 2010; Kunert et al., 2012; Meißner et al., 2012; Forrester, 2014; Lübbe et al., 2016).

Briefly, throughfall and soil water variability, soil properties, and root water uptake patterns form complex and intertwined interactions in the terrestrial hydrological cycle. It has not yet been shown empirically how root water uptake patterns are affected by throughfall and soil water variation in combination with soil properties and neighboring tree characteristics. Therefore, here we investigate the role of throughfall patterns and pose the following questions to guide the investigation:

i) How do throughfall patterns influence root water uptake patterns?
ii) How do soil water and its variation and soil properties control variation in root water uptake?
iii) What is the role of biotic factors, namely tree size, distance, number, and species richness, on root water uptake patterns?

Here, we address these questions by employing linear mixed effects model based on weekly throughfall sampling at locations paired with intensive soil moisture measurements in a beech-dominated unmanaged forest. In addition, we incorporate data on field capacity, bulk density, and neighboring tree characteristics.

# 2) Materials and Methods

## 2.1) Research Site and Field Sampling

### 2.1.1) Research Site

The research site is located in the forested upper hill region of the Hainich low mountain range in Thuringia, Germany, as a part of the Hainich Critical Zone Exploratory (CZE) (Küsel et al., 2016). The

altitude in the research site ranges from 362 m to 368 m a.s.l. Mean annual air temperature varies between 7.5 and 9.5 °C, and the mean annual precipitation ranges from less than 600 to 1000 mm in the CZE (Küsel et al., 2016).

In the study area, thin-bedded alternations of limestones and marlstones of carbonate rock (Middle Triassic) form the bedrock overlain by shallow Pleistocene loess layer with cambisols and luvisols as dominant soil types (IUSS Working Group, 2006; Metzger et al., 2021). The median soil depth above the weathered bedrock is 37 cm, with soil depths ranging from 15 cm to a maximum depth of 87 cm (Metzger et al., 2017).

In 2019, the tree community in the research site consisted of 574 individuals of various ages (diameter at breast height ≥ 5cm). The dominant species is European beech (*Fagus sylvatica* L.), which makes up 70% of the tree community, followed by sycamore maple (*Acer pseudoplatanus* L.) with 21 %, and European ash (*Fraxinus excelsior* L.) with 4%. These dominant species are accompanied by Large-leaved linden (*Tilia platyphyllos* Scop.), European hornbeam (*Carpinus betulus* L.), Norway maple (*Acer platanoides* L.), Scots elm (*Ulmus glabra* L.), and Wild service tree (*Sorbus torminalis* (L.) Crantz). The stand has a total basal area of 40 $m^2$ $ha^{-1}$ and has been unmanaged since 1997 (Kohlhepp et al., 2017).

### 2.1.2) Soil moisture monitoring and soil properties

The forest site (1 ha) was equipped with a soil moisture monitoring network (SoilNet; Bogena et al., 2010) consisting of SMT100 frequency domain sensors (Treuebner GmbH, Neustadt, Germany). Metzger et al. (2017) first described the soil moisture monitoring setup. Briefly, the observation platform (Figure 1) was divided into 100 subplots (10 m × 10 m), and 49 subplots were equipped with soil moisture sensors at two random measuring points each, for a total of 98 locations. At each measuring point, sensors were placed at two different depths, 7.5 cm (top sensors) and 27.5 cm (bottom sensors). The soil moisture network is maintained through a regular bi-weekly routine to avoid potential failures such as depleted sensors batteries, hardware problems, etc.

Undisturbed soil samples were collected during the sensor installation in 2014 and 2015 to estimate bulk density and water content at field capacity. In addition, we collected additional disturbed soil samples (n = 40) near sensor locations in 2019. Bulk density was determined from oven-dried (24h, 105°C) soil mass

weight and water content at field capacity by applying 60 hPa pressure to the saturated undisturbed sample for 72 h.

Soil properties vary slightly from top to subsoil at the research site. While silty loam is the dominant soil texture in both layers, the clay content is higher in the subsoil (Metzger et al., 2021). The median volumetric water content at the field capacity is 44% in the topsoil and 42% in the subsoil. Moreover, the water content at the field capacity varies from 27% to 60% and from 31% to 62% in the topsoil and subsoil, respectively. The average bulk density ($d_{bulk}$) of the topsoil is 1.16 g $cm^{-3}$, with a range of 0.73 to 1.5 g $cm^{-3}$. In the subsoil, the average bulk density ($d_{bulk}$) is slightly higher at 1.37 g $cm^{-3}$ but has a similar range (0.7 - 1.6 g $cm^{-3}$) (See supplement for details).

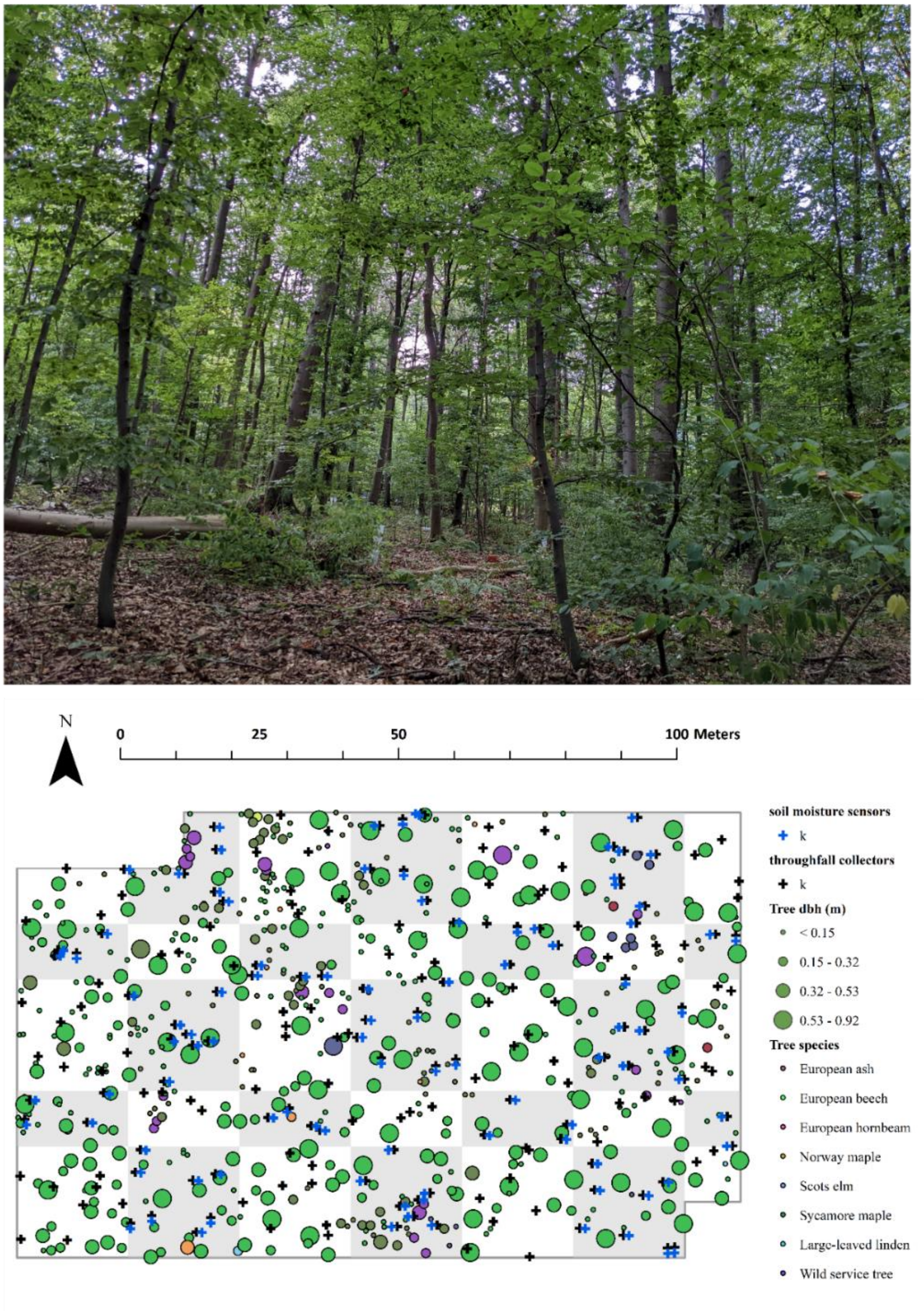

**Figure 1** (above) The photo of the site. (below) the field monitoring setup of stratified randomly distributed throughfall collectors and soil moisture sensors together with the trees which are sized according to the diameter at breast height (dbh) and coloured according to the species. Throughfall collectors are paired with soil moisture sensors at 98 locations (n=182) in the grey shaded subplots. White coloured subplots are equipped with only throughfall collectors.

### 2.1.3) Gross precipitation and throughfall sampling

Five gross precipitation funnels were placed 1.5 m above ground level in an adjacent open grassland (ca. 250 m distance to the research site). As described in Metzger et al. (2017) and Demir et al. (2022), the precipitation funnels were made of a circular plastic funnel (12 cm in diameter) and sampling bottle (2 L in volume), and ping pong balls were placed in the funnel orifice to prevent evaporation losses.

During the early growing season of 2019, we placed throughfall collectors in soil moisture monitoring subplots at 98 locations. We paired these throughfall collectors with the soil moisture sensors by placing them within 1 m of each other. The paired collectors were placed down-slope to avoid interference with soil moisture measurements. For the rest of the research site, in 51 other subplots, we adopted a separate independent stratified random design from Metzger et al. (2017). Briefly, we placed two throughfall collectors in each subplot that was not equipped with soil moisture sensors. All throughfall collectors were placed roughly 37 cm above the ground.

We conducted weekly manual measurement of throughfall and gross precipitation during the 2019 growing season (April to August). We measured gross precipitation and throughfall on rainless days therefore, in some of the sampling weeks, the interval between field measurements ranged between six and eight days.

We used the paired throughfall collectors (n = 98) to identify the drivers of root water uptake patterns as we derived root water uptake values based on soil water content measurements (see below). However, we used all randomly placed throughfall collectors (n = 200) to describe the spatio-temporal variation of throughfall within the research site.

## 2.2) Estimation of potential evapotranspiration

We calculated the daily potential evapotranspiration by applying the concept of thermodynamic limits of convection (Kleidon and Renner, 2013):

$$E_{pot}=\frac{1}{\lambda}\frac{s}{s+\gamma}\frac{R_{sn}}{2} \quad (1)$$

Where $R_{sn}$ is absorbed solar radiation (W m$^{-2}$), $\lambda$ is the latent heat of vaporization, $\gamma$ is the psychrometric constant, and $s$ is the slope of the saturation vapor pressure curve.

Here, we acquired solar radiation, air temperature, and precipitation data for the throughfall sampling period from a nearby weather station ("Reckenbuel") which is located approximately 1.4 km northeast of the research site and provides data in 10 minutes intervals. The site-specific albedo for the summer period was adopted from Otto et al. (2014).

In addition, we used the precipitation data measured at the weather station to define rain events and dry periods, as described below.

## 2.3) Data analysis

### 2.3.1) Quality control of soil water content data

We systematically reviewed the six-minute soil water content data for quality control in two steps: 1) identification of problems (such as jumps to extremely low and high values, duplicated time stamps of different values, long discontinuities in the measurements, and lack of temporal variation in the time series despite rain events), 2) classification and removal of detected outliers and irregularities. We visually identified and removed unrealistic measurements such as extremely low (< 5 vol-%) and high values far beyond the field capacity (> 75 vol-%) and long plateaus of repeated values despite rain events. We also excluded the time series that exhibited long-term discontinuities that prevented us from calculating root water uptake. During the visual inspection, we eliminated values with duplicated time stamps that violated the actual temporal trend. Next, we scanned the data using the Hampel filter function of the 'pracma' R package (Borchers, 2021) with customized moving window length and Pearson's rule threshold value (Pearson, 1999) to flag possible outliers.

Despite regular maintenance, many sensors failed to meet the quality criteria in the growing season (March-August) in 2019. Only 56 sensor locations (out of 98) simultaneously provided high-quality data from both top and bottom sensors with different time intervals throughout the season. Of these, only 34 sensor locations provided data for the root water uptake estimation.

### 2.3.2) Soil water calculation

We estimated soil water (S) at measurement locations for the monitored soil layer based on volumetric soil water content measured by top and bottom sensors.

$$S_{i,d} = \sum z_t \theta^t_{i,d} + z_b\ \theta^b_{i,d} \quad (2)$$

We similarly integrated the soil water at field capacity ($S_{FC,i}$,)

$$S_{FC,i} = \sum z_t \theta^t_{FC,i} + z_b\ \theta^b_{FC,i} \quad (3)$$

where $z_t$ is the depth of the soil column monitored by the top sensor and $z_b$ is the depth of soil represented by the bottom sensor, and $\theta_{i,d}$ is a volumetric soil water content at location *i* on date *d*, and $\theta_{FC,i}$ the soil water content at the field capacity.

We calculated bulk density at the sensors' locations for the monitored soil layer.

$$\overline{d_{bulk,i}} = \frac{\sum z_t d^t_{bulk,i} + z_b\ d^b_{bulk,i}}{\sum z_t + z_b} \quad (4)$$

where $d^t_{bulk,i}$ and $d^b_{bulk,i}$ are the bulk density of the topsoil and subsoil, respectively, at location *i*.

### 2.3.3) Descriptive Statistics

We calculated the coefficient of quartile variation (CQV) and the interquartile range to describe spatial variation of throughfall, volumetric soil water content, and root water uptake. Also, we estimated octile skewness ($OS_8$) of throughfall based on the first and seventh octile and standard deviation (SD) of the estimated daily root water uptake.

$$CQV = \frac{Q_3 - Q_1}{Q_3 + Q_1} \quad (5)$$

$$OS_8 = \frac{(Q_7 - median) - (median - Q_1)}{Q_7 - Q_1} \quad (6)$$

We characterized spatial patterns of daily root water uptake ($E_t$) by calculating the spatial deviation from the mean ($\delta E_{t\ i,d}$, Equation 7) (Vachaud et al., 1985).

$$\delta E_{t\ i,d} = \frac{E_{t,\ i,d} - \overline{E_{t,d}}}{\overline{E_{t,d}}} \quad (7)$$

where $E_{t,\ i,d}$ is daily root water uptake estimated at *i* sensor location on date *d* and $\overline{E_{t,d}}$ is spatial average of daily root water uptake on date *d*.

Similarly, we calculated the spatial deviation of soil water and throughfall to identify their spatial patterns.

## 2.4) Root water uptake estimation

We estimated root water uptake using the multi-step, multi-layer regression method (MSML), which derives evapotranspiration from diurnal differences in soil water content (Guderle and Hildebrandt, 2015; Guderle et al., 2018). This approach does not require prior information on root structure but relies on high temporal and spatial resolution data on multiple soil layers.

As described in Guderle and Hildebrandt (2015), the MSML derives root water uptake from distinct differences in the day and night portions of soil moisture time series. The main assumption is that in the absence of rainfall-driven rapid vertical soil water flow, evapotranspiration occurs only during the day, while soil water flow occurs both during the day and at night. As a result, soil moisture time series reflect a distinct day/night signal under dry weather conditions. This method has previously been applied to estimate transpiration in both forest and grassland ecosystems (Guderle et al., 2018; Jackisch et al., 2020). Therefore, we first excluded potential periods of fast vertical flow periods from the time series due to previous rainfall events and identified periods for estimating daily root water uptake. We considered 8 h buffer period to include canopy dripping and 48 h for the cessation of rainfall influence on soil water. Thus, a total of 56 h was the time interval used to define the water uptake estimation period. The period when the root water uptake is estimated is hereafter referred to as the dry period.

Next, we split each soil moisture time series into a day (transpiration active period) and a night branch, as Guderle and Hildebrandt (2015) explained. We defined the transpiration period (starts 2 h after sunrise and ends 2 h before sunset) based on local sunrise and sunset time. Sunrise and sunset times were obtained from the R package 'suncal' (Thieurmel and Elmarhraoui, 2022). We fit linear models to each split branch of the time series and derived the slopes. The difference between the slope of the day branch ($m_{tot}$) and the average slope of the antecedent and preceding night ($\overline{m_{flow,i}}$) gives the rate of water uptake. Thus, we estimated daily evapotranspiration at each soil water content location *i* (Equation 8, 9) by accounting for soil layer thickness and slope difference.

$$E_{t,msml,i}^{t,b} = (m_{tot,i}^{t,b} - \overline{m_{flow,i}^{t,b}})\, d_{z,i}^{t,b}{}_{z,i} \quad (8)$$

$$E_{t,i} = \sum(E_{t,msml,i}^{t} + E_{t,msml,i}^{b}) \quad (9)$$

## 2.5) Linear Mixed Effects Model

We employed a linear mixed effects model to investigate the driving factors for root water uptake patterns. A linear mixed effects model is a multivariate statistical tool. It describes the relationship between a dependent variable and explanatory variables (fixed effects) while controlling for dependencies in the data that may arise due to repeated sampling with certain designs (random effects).

For the model, we used only paired throughfall and soil moisture measurement locations where both top and bottom sensors provided data within the dry periods. All considered potential controlling factors for root water uptake patterns are listed in Table 1. These are daily spatial average soil water storage, the spatial deviation of soil water from the mean, soil water at field capacity and bulk density of the monitored soil layer, number of trees, and number of species within a 5 m radius of each soil moisture location, and inverse distanced basal area (BA) within 5 m radius of each soil moisture location. Basal area was calculated as follows:

$$BA_i = \frac{\sum_{R=1}^{R} W_R A_{tree}}{A} \tag{10}$$

with $$W_R = \frac{(x_i - x_R)^2}{\sum_R (x_i - x_R)^2} \tag{11}$$

where i is the soil moisture sensor located at $x_i$, R is the tree index located at $x_R$, and $A_{tree}$ is the individual basal area of the corresponding tree, A is the area around the soil moisture sensor i with 5 m in radius.

Even though our research plot is a beech-dominated forest, in some spots, two to four species were present within a 5 m radius of the soil moisture sensors.

Moreover, we quantified the spatial variability of throughfall as the difference between the throughfall measured at a given location and the spatial mean normalized by the spatial mean. Here we considered this variable at a two-time scales: the week(s) prior to root water uptake estimation period, and the median of the entire measurement period. We also included interaction terms (Table 1) as fixed factors in the model. Because of repeated observations at the measurement locations, soil moisture sensor points and dry periods, (i.e., the root water uptake estimated time interval), were considered as random effects.

We conducted all analyses with the R statistical software (R Core Team, 2022) and used the *lmer* function in the 'lme4' package (Bates et al., 2015) for the model development. We visually checked the model assumptions using the 'check_model' function of the 'performance' package (Lüdecke et al., 2021).

In addition, we calculated both conditional and marginal $R^2$ of the model with the 'MuMIn' package (Bartoń, 2020). While the conditional $R^2$ includes the variance of the entire model, the marginal $R^2$ subsumes only the fixed effects (Bartoń, 2020). Before fitting the linear mixed effects model, we tested for co-linearity of the considered variables and scaled the data with a Z-transformation by using the 'scale' function in base R (R Core Team, 2022), which allowed us to evaluate the individual effect of fixed effects by comparing slopes and significance levels.

We developed the optimal model by applying a systematic model selection procedure based on Akaike's Information Criterion (AIC) comparison in combination with the examination of the factors. Model selection began with the beyond-optimal model, which included all possible fixed and random effects. We stepwise evaluated each fixed effect based on its respective significance ($p$ value comparision) by fitting the model the maximum likelihood (ML) to be able to compare AIC values (Zuur et al., 2009). In each step, starting with interaction terms, we identified the least significant effect and formulated a model without it. We compared the AIC values of the model before and after removing the effect, discarding it in case the AIC was unaffected or decreased. We followed the procedure with the next equally detected effect, and repeated it until only significant fixed effects remained, and the model with the lowest AIC (the optimal model) was obtained.

As a final step, the best model was refitted with restricted maximum likelihood (REML) (Zuur et al., 2009).

**Table 1** List of fixed and random factors considered for estimating the root water uptake patterns through linear mixed effects model. Interaction is shown with 'x'.

| **Fixed Factors** | |
|---|---|
| **Single Factors** | **Interaction Factors** |
| Spatial average of soil water storage in the monitored soil layer ($\bar{S}$) | $\bar{S} \times S_{FC}$ |
| Spatial deviation of soil water storage from the mean ($\delta S$) | $\delta S \times S_{FC}$ |
| Field capacity of the monitored soil layer ($S_{FC}$) | $\delta S \times BA$ |
| Bulk density capacity of the monitored soil layer ($d_{bulk}$) | $\bar{S} \times BA$ |
| Spatial deviation of throughfall of events measured in sampling week previous to the corresponding dry period ($\delta P_{TF_{last\ ev.}}$) | $\delta S \times n_{tree}$ |
| The median of spatial deviation of throughfall measured within the whole sampling period ($\widetilde{\delta P_{TF}}$) | $\bar{S} \times n_{tree}$ |
| Number of trees ($n_{tree}$) | $\delta P_{TF_{last\ ev.}} \times S_{FC}$ |
| Basal area (BA) | $\delta P_{TF_{temp.\ stable.}} \times S_{FC}$ |
| Number of species ($n_{sp,tree}$) | $\delta P_{TF_{last\ ev.}} \times d_{bulk}$ |
| | $\delta P_{TF_{temp.\ stable.}} \times d_{bulk}$ |
| | $n_{sp,tree} \times WA_{int}$ |
| **Random factors** | |
| Soil moisture sensor location | |
| Dry period | |

# 3) Results

## 3.1) Spatio-temporal distribution of throughfall and soil water content

In 12 out of the 16 sampling weeks, the weekly gross precipitation was more than half of the total potential evapotranspiration. Table 2 further shows the distribution of throughfall sampled in 2019 (April-August) at 200 collectors and the 98 collectors that were paired with soil moisture sensors. The weekly throughfall increased with the increase in rain events. Additionally, the coefficient of quartile variation (CQV) of
throughfall was generally lower for larger cumulative weekly rains. On average, the paired collectors received similar amounts of throughfall to all collectors (Table 2). The CQV of data from the paired collectors ranged from 0.27 to 0.6, which is similar to the CQV of throughfall sampled at all collectors. The octile skew ($OS_8$) of paired and of all collectors was also similar.

As the growing season progressed in 2019, the average soil water content decreased in both the topsoil and subsoil. In April and early May, the average volumetric soil water content in the top soil was above 30%, which dropped to below 10% by the end of August. In the subsoil, the volumetric soil water content similarly declined from above 40 % to below 20 % over the sampling period (Figure 2). On average, soil water changed from 52.5mm to 17.5 mm in the topsoil and from 80 mm to 40mm in the subsoil.

We derived root water uptake for four periods (19 days) under different soil wetness conditions that captured the seasonal variation of soil water content, including late spring when the soil water content was higher, following re-wetting with late summer rains. As listed in Table 3 and shown in Figure 2, two periods were in late May and early June, and each lasted two days. The third period began in late June and lasted 11 days; the last was four days in late July. During these periods, the average soil water content declined from 33 to 15 % in the topsoil and from 43 to 27% in the subsoil. Table 3 additionally shows that within the dry periods, the coefficient of quartile variation (CQV) of soil water content was between 0.09 -0.14 and 0.08 to 0.16 in the topsoil and subsoil. During the dry periods, the spatial heterogeneity of soil water content in the subsoil increased systematically. In contrast, the spatial variation of topsoil soil water content did not correlate with soil dryness.

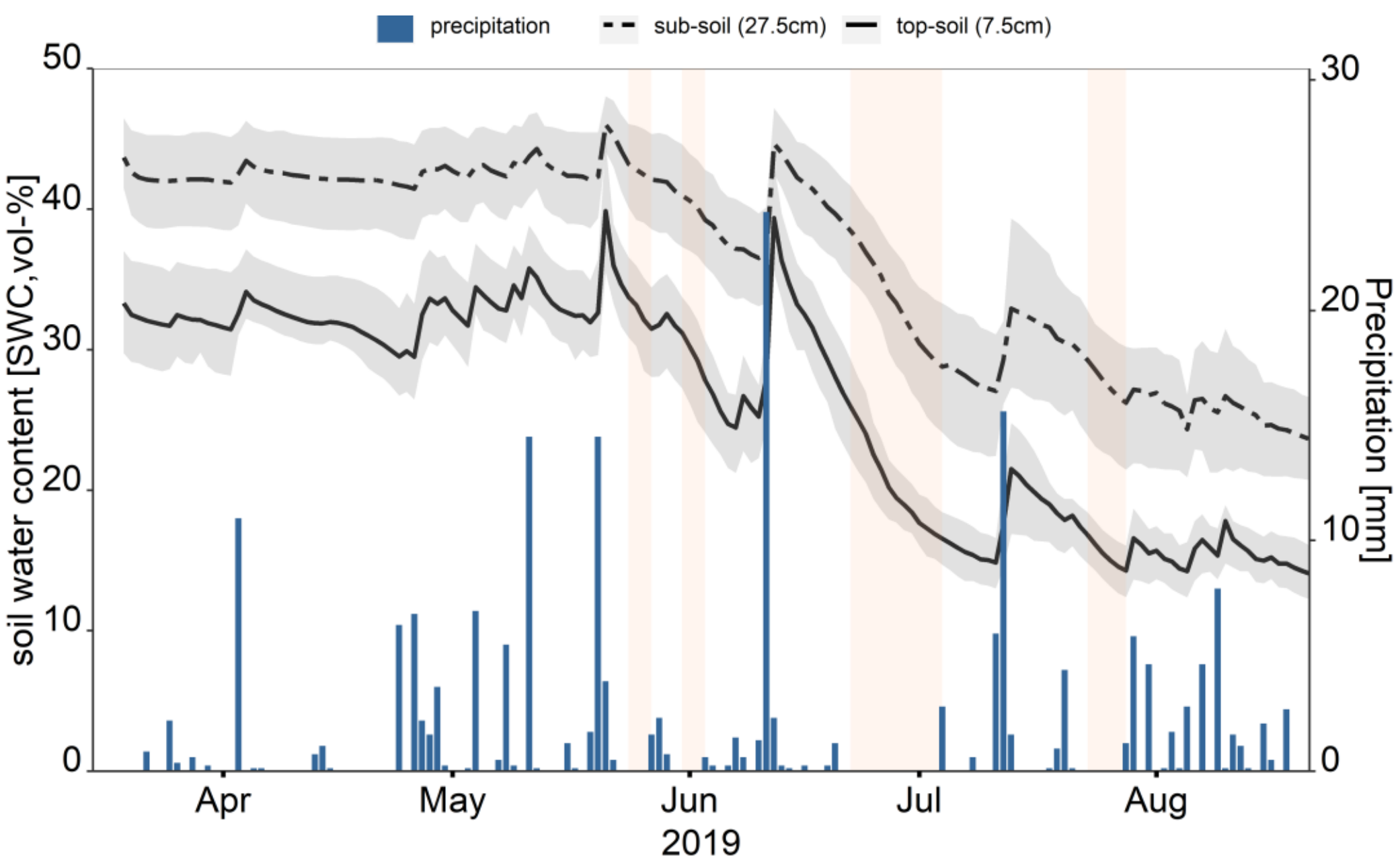

329
330 **Figure 2** Soil moisture temporal variation in top and subsoil together with the daily precipitation measured at the nearby
331 Reckenbühl station (approximately 1.4 km to the Northeast). The solid and dashed lines are spatial mean of soil water content
332 estimated based on top (7.5 cm) and bottom (27.5 cm) sensors, and grey shaded areas show first and third quartiles. The reddish
333 shaded areas show defined dry periods within the throughfall sampling when root water uptake could be estimated.

**Table 2** Cumulative potential evapotranspiration in mm ($E_{pot,cum}$), gross precipitation ($P_g$), the ratio of total precipitation to the potential evapotranspiration, spatial mean of throughfall based on all collectors ($\overline{P_{TF}}$), spatial mean of throughfall based paired collectors ($\overline{P_{TF}}$) in mm, interquartile range (IQR), coefficient of quartile variation (CQV) and octile skewness ($OS_8$) of both all and paired throughfall collectors during the sampling week. The values are ordered according to the cumulated gross precipitation size.

| Date | $E_{pot,cum}$ | $P_g$ | $P_g/E_{pot}$ | $\overline{P_{TF}}$ | $\frac{IQR}{\overline{P_{TF}}}$ | $\frac{CQV}{\overline{P_{TF}}}$ | $\frac{OS_8}{\overline{P_{TF}}}$ | $\overline{P_{TF}}_{paired}$ | $\frac{IQR}{\overline{P_{TF}}_{paired}}$ | $\frac{CQV}{\overline{P_{TF}}_{paired}}$ | $\frac{OS_8}{\overline{P_{TF}}_{paired}}$ |
|---|---|---|---|---|---|---|---|---|---|---|---|
| 04-06-2019 | 13.55 | 0.76 | 0.06 | 0.35 | 0.18 | 0.25 | 0.46 | 0.34 | 0.16 | 0.24 | 0.49 |
| 26-06-2019 | 20.87 | 1.73 | 0.08 | 0.97 | 0.44 | 0.24 | 0.16 | 0.98 | 0.53 | 0.27 | 0.27 |
| 17-04-2019 | 5.62 | 2.42 | 0.43 | 1.72 | 0.27 | 0.08 | 0.23 | 1.72 | 0.33 | 0.09 | 0.09 |
| 18-06-2019 | 9.46 | 4.00 | 0.42 | 2.58 | 0.62 | 0.12 | -0.03 | 2.57 | 0.53 | 0.10 | -0.08 |
| 29-05-2019 | 10.15 | 6.27 | 0.62 | 3.77 | 1.24 | 0.17 | -0.52 | 3.63 | 1.50 | 0.21 | -0.42 |
| 24-07-2019 | 13.52 | 7.80 | 0.58 | 4.61 | 1.06 | 0.12 | -0.34 | 4.48 | 0.88 | 0.10 | -0.63 |
| 21-08-2019 | 8.94 | 8.54 | 0.96 | 5.19 | 1.06 | 0.10 | -0.47 | 5.17 | 0.97 | 0.10 | -0.44 |
| 30-07-2019 | 12.68 | 10.73 | 0.85 | 7.81 | 2.25 | 0.15 | -1.51 | 7.58 | 2.28 | 0.15 | -1.17 |
| 07-05-2019 | 6.65 | 12.56 | 1.89 | 9.21 | 1.33 | 0.07 | -0.75 | 9.21 | 1.99 | 0.11 | -1.05 |
| 14-08-2019 | 8.51 | 13.79 | 1.62 | 11.19 | 2.65 | 0.12 | -1.40 | 10.99 | 2.98 | 0.13 | -1.13 |
| 08-08-2019 | 13.91 | 23.87 | 1.72 | 16.60 | 2.65 | 0.08 | -1.10 | 16.52 | 2.65 | 0.08 | -1.17 |
| 30-04-2019 | 5.93 | 24.47 | 4.13 | 18.44 | 3.09 | 0.08 | -1.63 | 18.30 | 2.65 | 0.07 | -1.23 |
| 17-07-2019 | 8.28 | 29.27 | 3.54 | 24.22 | 3.54 | 0.07 | -2.08 | 24.39 | 3.54 | 0.07 | -2.59 |
| 15-05-2019 | 7.42 | 29.53 | 3.98 | 22.10 | 3.54 | 0.08 | -2.11 | 22.21 | 3.54 | 0.08 | -2.11 |
| 22-05-2019 | 6.74 | 41.82 | 6.20 | 30.94 | 3.54 | 0.06 | -3.04 | 30.54 | 3.54 | 0.06 | -3.46 |
| 13-06-2019 | 14.47 | 71.84 | 4.96 | 57.77 | 8.51 | 0.07 | -5.82 | 57.99 | 7.29 | 0.06 | -6.52 |

**Table 3** The spatial average of daily volumetric soil water content ($\overline{\theta_{top\text{-}soil}}$, vol-%) in topsoil (0-17.5 cm), and ($\overline{\theta_{subsoil}}$, vol-%) in subsoil (17.5 – 37.5 cm) during the defined dry periods. The inter quartile range (IQR), and coefficient of quartile variation (CQV) of daily volumetric soil water content in both layers during the dry periods.

| Date | $\overline{\theta_{top\text{-}soil}}$ (vol-%) | IQR $\theta_{top\text{-}soil}$ (vol-%) | CQV $\theta_{top\text{-}soil}$ (vol-%) | $\overline{\theta_{sub-soil}}$ (vol-%) | IQR $\theta_{subsoil}$ (vol-%) | CQV $\theta_{subsoil}$ (vol-%) | Dry Period |
|---|---|---|---|---|---|---|---|
| 25 -05-2019 | 33.17 | 5.72 | 0.09 | 42.82 | 6.72 | 0.08 | 1 |
| 26-05-2019 | 32.12 | 6.62 | 0.10 | 42.46 | 6.67 | 0.08 | 1 |
| 01-06-2019 | 30.23 | 6.87 | 0.12 | 40.61 | 6.9 | 0.09 | 2 |
| 02-06-2019 | 29.22 | 7.23 | 0.13 | 40.11 | 6.85 | 0.09 | 2 |
| 23-06-2019 | 25.01 | 6.69 | 0.14 | 37.80 | 6.38 | 0.08 | 3 |
| 24-06-2019 | 24.04 | 6.45 | 0.14 | 36.94 | 6.22 | 0.08 | 3 |
| 25-06-2019 | 22.52 | 5.43 | 0.12 | 36.13 | 6.54 | 0.09 | 3 |
| 26-06-2019 | 21.48 | 5.07 | 0.12 | 35.24 | 6.71 | 0.10 | 3 |
| 27-06-2019 | 20.20 | 4.25 | 0.11 | 33.98 | 7.75 | 0.12 | 3 |
| 28-06-2019 | 19.45 | 3.85 | 0.10 | 33.31 | 8.08 | 0.12 | 3 |
| 29-06-2019 | 18.98 | 3.83 | 0.10 | 32.36 | 8.05 | 0.12 | 3 |
| 30-06-2019 | 18.44 | 3.52 | 0.09 | 31.37 | 8.15 | 0.13 | 3 |
| 01-07-2019 | 17.67 | 3.62 | 0.10 | 30.45 | 8.18 | 0.13 | 3 |
| 02-07-2019 | 17.29 | 4.18 | 0.12 | 29.84 | 8.87 | 0.15 | 3 |
| 03-07-2019 | 16.89 | 3.72 | 0.11 | 29.26 | 8.98 | 0.15 | 3 |
| 24-07-2019 | 16.15 | 3.48 | 0.11 | 28.56 | 8.7 | 0.16 | 4 |
| 25-07-2019 | 15.51 | 3.47 | 0.11 | 27.85 | 8.67 | 0.16 | 4 |
| 26-07-2019 | 14.98 | 3.57 | 0.12 | 27.21 | 8.49 | 0.16 | 4 |
| 27-07-2019 | 14.57 | 3.65 | 0.13 | 26.65 | 8.63 | 0.16 | 4 |

## 3.2) Soil water storage, potential evapotranspiration, and root water uptake

The integrated field capacity of the monitored soil depth was 160 mm on average at the research site. Table 4 shows that soil water storage was much lower than the field capacity during the dry periods, and the mean soil water storage dropped below 42 mm in late July. In addition, Table 4 demonstrates that the average root water uptake ($\overline{E_t}$) ranged from 0.94 mm $d^{-1}$ to 3 mm $d^{-1}$ while potential evapotranspiration ($E_{pot}$) ranged from 1.75 mm $d^{-1}$ to 3.12 mm $d^{-1}$. The discrepancy between average root water uptake and the potential evapotranspiration increased as soil water storage assessed by the soil sensors progressively decreased, especially during the longest dry period (Table 4). Root water uptake showed greater spatial variation than water input and soil wetness. The coefficient of quartile variation (CQV) of root water uptake ranged from 0.15 to 0.28, which was higher than the CQV of throughfall and volumetric soil water content in both soil layers.

**Table 4** The daily average air temperature ($T_{air}$ ,°C), potential evapotranspiration ($E_{pot}$, mm), mean soil water storage ($\bar{S}$, mm) in monitored soil layer (0 - 37.5 cm), and spatial mean of daily root water uptake ($\overline{E_t}$ , mm) based on all soil moisture sensors, and the ratio of the root water uptake to the potential evapotranspiration together with and standard deviation (SD) and coefficient of quartile variation (CQV) of the daily root water uptake during the defined dry periods

| Date | $T_{air}$ (°C) | $E_{pot}$ (mm) | $\bar{S}$ (mm) | $\overline{E_t}$ (mm) | $\overline{E_t}/E_{pot}$ (%) | SD $\overline{E_t}$ | CQV $\overline{E_t}$ | Dry Period |
|---|---|---|---|---|---|---|---|---|
| 25-05-2019 | 12.74 | 1.80 | 71.94 | 1.09 | 60.56 | 0.38 | 0.28 | 1 |
| 26-05-2019 | 14.43 | 1.90 | 70.57 | 1.30 | 68.42 | 0.48 | 0.25 | 1 |
| 01-06-2019 | 18.42 | 2.59 | 67.16 | 2.26 | 87.26 | 0.98 | 0.27 | 2 |
| 02-06-2019 | 21.38 | 2.77 | 65.79 | 2.50 | 90.25 | 1.12 | 0.18 | 2 |
| 23-06-2019 | 19.45 | 2.79 | 59.81 | 2.83 | 101.43 | 0.90 | 0.19 | 3 |
| 24-06-2019 | 20.22 | 2.82 | 58.16 | 2.62 | 92.91 | 0.76 | 0.17 | 3 |
| 25-06-2019 | 22.52 | 2.89 | 55.96 | 2.67 | 92.39 | 0.78 | 0.16 | 3 |
| 26-06-2019 | 25.73 | 2.96 | 54.13 | 3.00 | 101.35 | 0.88 | 0.15 | 3 |
| 27-06-2019 | 18.83 | 2.75 | 51.91 | 2.28 | 82.91 | 0.55 | 0.16 | 3 |
| 28-06-2019 | 16.07 | 2.58 | 50.55 | 1.53 | 59.30 | 0.40 | 0.20 | 3 |
| 29-06-2019 | 19.59 | 2.85 | 49.55 | 2.11 | 74.04 | 0.60 | 0.20 | 3 |
| 30-06-2019 | 25.54 | 3.12 | 48.26 | 2.57 | 82.37 | 0.86 | 0.18 | 3 |
| 01-07-2019 | 20.63 | 2.30 | 46.69 | 1.59 | 69.13 | 0.53 | 0.18 | 3 |
| 02-07-2019 | 14.88 | 1.75 | 45.81 | 1.08 | 61.71 | 0.42 | 0.24 | 3 |
| 03-07-2019 | 13.77 | 1.91 | 44.95 | 0.94 | 49.21 | 0.30 | 0.23 | 3 |
| 24-07-2019 | 24.39 | 2.76 | 43.61 | 1.88 | 68.12 | 0.64 | 0.19 | 4 |
| 25-07-2019 | 25.33 | 2.82 | 42.31 | 1.77 | 62.77 | 0.60 | 0.24 | 4 |
| 2019-07-26 | 23.27 | 2.64 | 41.18 | 1.40 | 53.03 | 0.55 | 0.18 | 4 |
| 2019-07-27 | 21.29 | 2.68 | 40.23 | 1.21 | 45.15 | 0.47 | 0.19 | 4 |

## 3.3) Soil water, throughfall, and root water uptake patterns

At soil moisture measurement points where daily root water uptake was determined (n = 34), we calculated the spatial deviation from the median of throughfall, soil water storage, and root water uptake to illustrate the spatial patterns. Figure 3 separately shows that some locations received repeatedly less (or more) throughfall than average ($\delta P_{TF} < 0$) throughout the sampling season. Similarly, some locations, either stored less water in the soil, i.e., were drier ($\delta S < 0$), and some places had lower root water uptake ($\delta E_t$) or higher than average water uptake throughout the sampling period. However, these locations were not related to each other. In fact, Figure 3 demonstrates that neither throughfall nor soil water patterns are directly correlated with the root water uptake patterns. For example, the locations with higher water uptake were not coupled with elevated throughfall input (locations colored dark) or higher soil water storage. In addition, soil water storage patterns were not correlated with throughfall patterns.

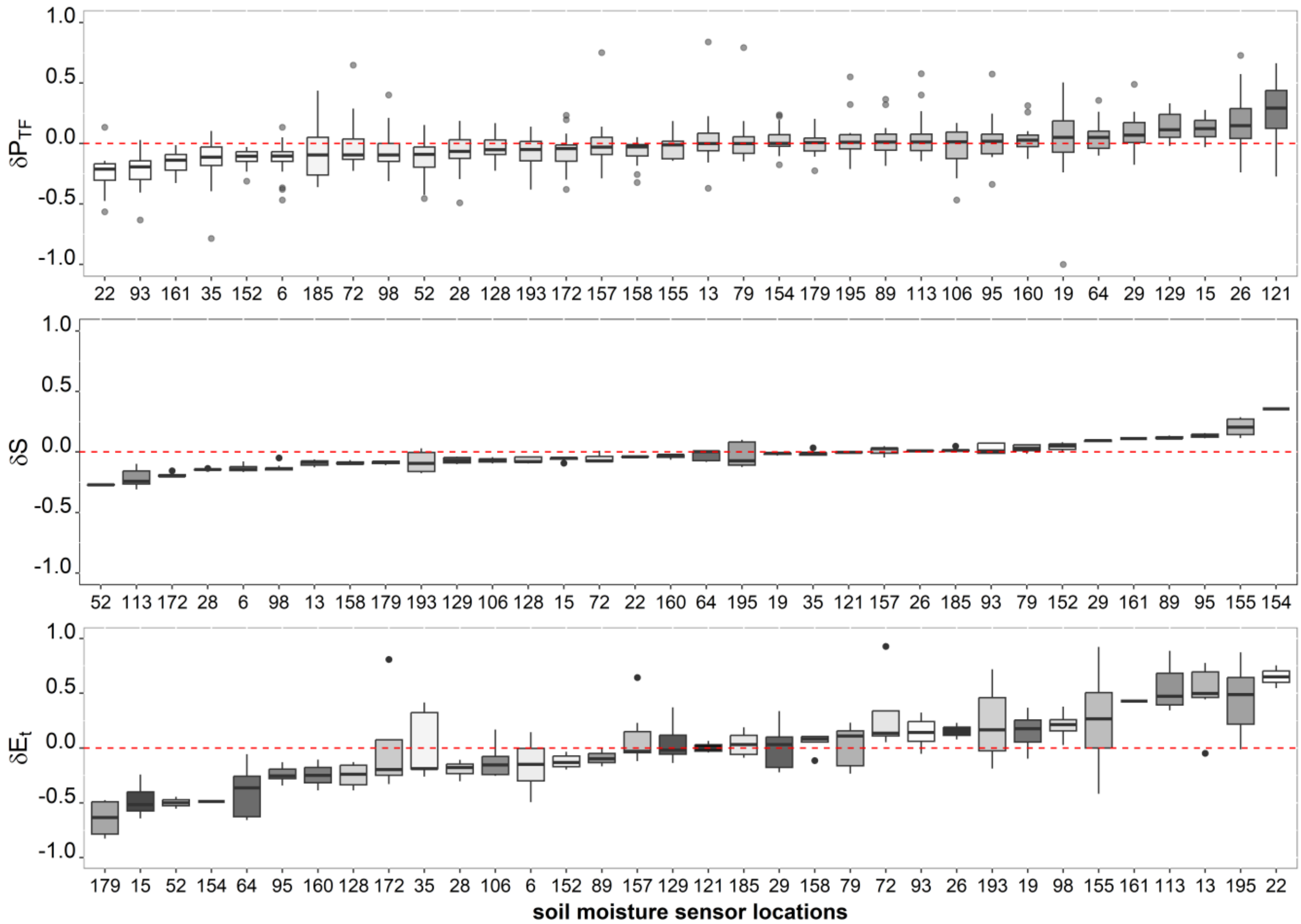

**Figure 3** Temporal stability of throughfall patterns which is estimated by the spatial deviation from the mean ($\delta P_{TF}$) throughout the sampling period in 2019 (April-August), soil water ($\delta S$) and root water uptake ($\delta E_t$) based on the spatial deviation from the mean during the defined dry periods. Soil moisture sensor locations colored according to throughfall input.

## 3.5) Fixed factors regulating root water uptake patterns

We used a linear mixed effects model to disentangle the effects of throughfall, soil water, soil properties, and the neighbouring tree characteristics on root water uptake patterns. The fixed and random effects contributed almost equally to the model. The $R^2$ of the model was 0.77, and the contribution of the fixed effect to the $R^2$ was 0.39 (See the supplement for more details on the optimal model).

Figure 4 shows only the significant fixed effects for root water uptake patterns. Spatial deviation of soil water from the mean (i.e., soil water patterns) was the only single and the most significant factor positively related to the spatial deviation of root water uptake. Thus, water uptake was elevated at locations where the most water was retained in the soil at the given time, i.e., greater soil water storage.

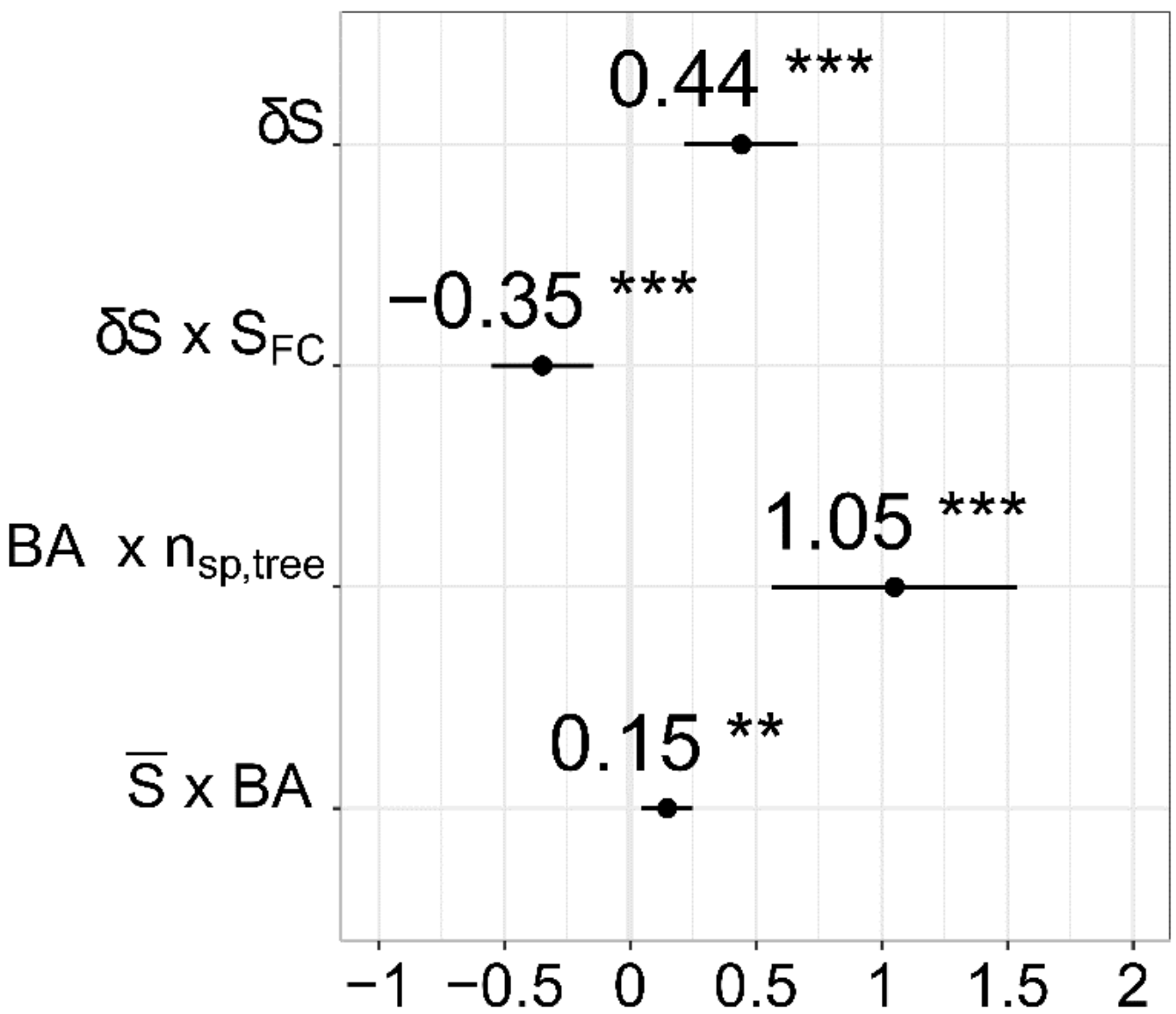

**Figure 4** The significant fixed factors of the best model to estimate root water uptake patterns ($\delta E_t$). Values on the x-axis indicate the slope of the relations. All variables were scaled by Z-transformation. Interaction is shown with 'x'. Here $\delta S$ is the spatial deviation of soil water, $S_{FC}$ is the field capacity, $n_{sp,tree}$ is the number of species, BA is the basal area, and $\bar{S}$ is soil water storage. Significance codes are *** ≅0, ** ≅ 0.001. (the details on the model can be found in the supplement)

Field capacity by itself was not a significant factor affecting local root water uptake. However, it strongly influenced how local soil water-controlled root water uptake as a part of the significant interaction term. Figure 5a illustrates how to root water uptake was more dependent on local soil water when field capacity was low (i.e., higher macroporosity). In contrast, soil bulk density and therefore total porosity was not part of the final model.

Although the spatial average of soil water storage, e.g., the state of wetness, was not an important factor for local root water uptake by itself, it moderated the impact of basal area (BA) on the spatial distribution of water uptake. We found that as the plot dries, uptake shifts from places with higher to places with lower basal area (Figure 5b). Furthermore, the statistical model revealed that water uptake increased with the higher basal area at locations where multiple species co-existed (Figure 5c). However, the number of species and the basal area were individually not significant fixed effects. Lastly, throughfall patterns were not significant predictors of local root water uptake. Only the median of the spatial deviation of

throughfall, which represents temporally stable patterns within the sampling period ($\widetilde{\delta P_{TF}}$), marginally improved the final model.

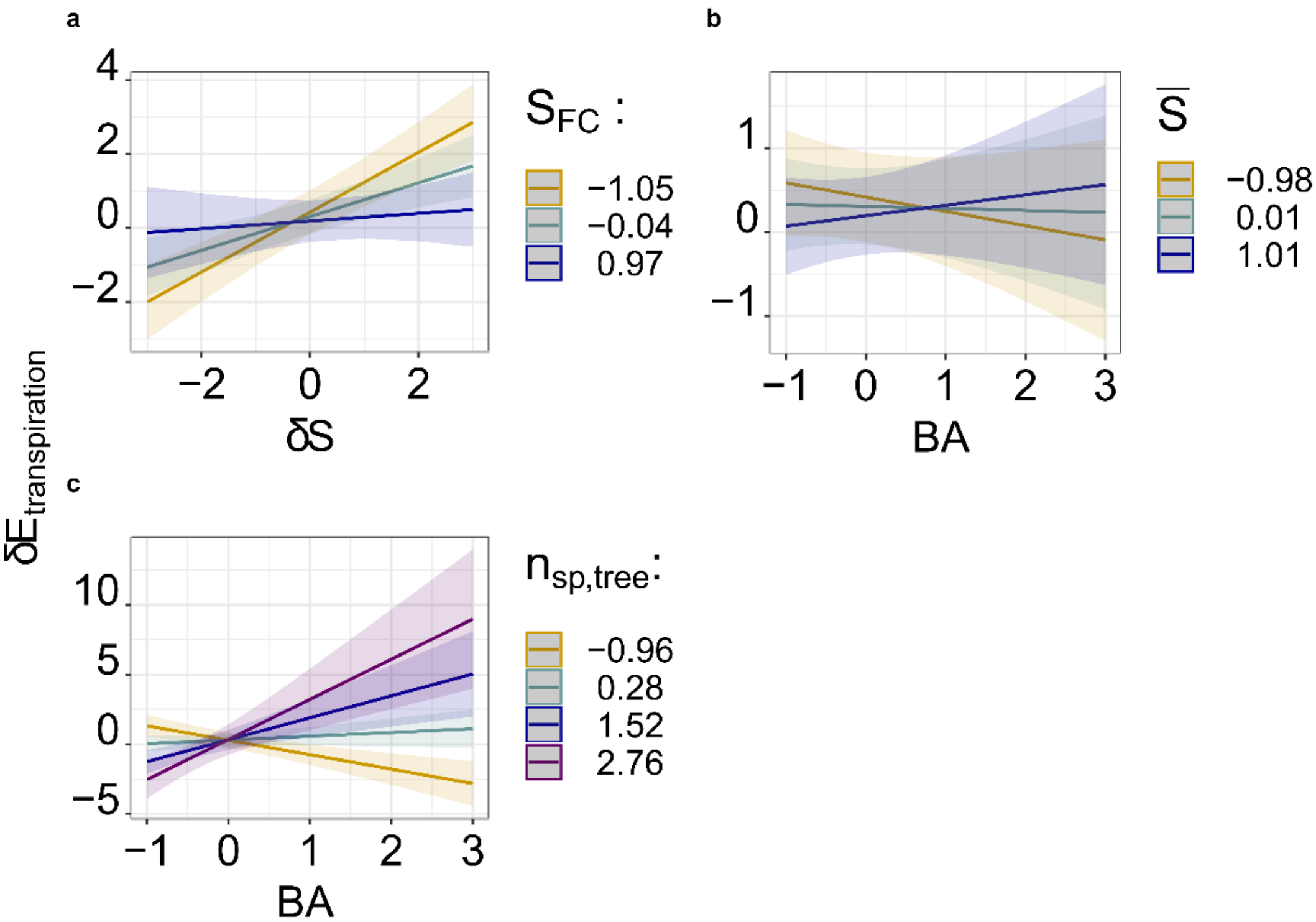

**Figure 5** Visualisation of the significant relations shown in Figure 4, representing the significant drivers of root water uptake patterns during the defined dry periods. Relation to (a) interactive relation of the spatial deviation of soil water storage and field capacity ($S_{FC}$), (b) the interactive relation of basal area (BA) and the spatial average of soil water storage ($\bar{S}$), (c) the interactive relation of number of species ($n_{sp,tree}$) and basal area (BA.).

# 4) Discussion

## 4.1) Spatial variation in throughfall does not affect root water uptake patterns

We adequately captured the spatial distribution and temporal stability of throughfall at locations where local root water uptake was derived. Consistent with previous observations in temperate forests (e.g., Whelan and Anderson, 1996; Staelens et al., 2006; Metzger et al., 2017), the amount of weekly rainfall significantly altered the spatial distribution of throughfall such that more rainfall, and thus more throughfall, resulted in less spatial variability. Previous studies repeatedly showed that throughfall patterns exhibit temporal stability in forest ecosystems (e.g., Keim et al., 2005; Staelens et al., 2006;

Wullaert et al., 2009; Rodrigues et al., 2022). At the same research site, using event-based sampling, Metzger et al., (2017) and Fischer et al., (2023) demonstrated that throughfall patterns persist over time, which was not different in our weekly sampling in 2019. With canopy cover being the key driver of throughfall (Fischer et al., 2023), it is not surprising that weekly cumulative events resulted in a localized high and low throughfall input.

Contrary to expectations (Bouten et al., 1992; Guswa and Spence, 2012; Coenders-Gerrits et al., 2013; Fischer et al., 2023), our results showed that throughfall hotspots do not increase or facilitate greater root water uptake. In addition, the linear mixed effects model results confirmed that throughfall patterns do not drive the variation in root water uptake. We attributed the absence of this to two reasons: (1) decoupled soil water and throughfall patterns, (2) non-water limited conditions.

Regarding (1), we confirmed that the temporally stable throughfall patterns do not correspond to the post-event soil water and root water uptake patterns. We paired the measurements of throughfall and soil water content measurements – and thus the estimates of root water uptake- within a distance of 1 m. The spatial correlation length of soil water content and throughfall is on the order of 6-10 m in natural temperate forests (Keim et al., 2005; Gerrits et al., 2010; Zehe et al., 2010). In the same study site with the spatially extended throughfall sampling, Fischer et al., (2023) found that the throughfall correlation length increased with decreasing event size, varying from 6.2 m to 9.5 m depending on the size of the rain events. Thus, the paired sampling design in our study likely provided co-located throughfall and soil moisture measurements. Nevertheless, only locations that stored more water than average rarely corresponded with the elevated throughfall input without a significant correlation. Hence, variation in soil water storage was not related to throughfall patterns despite temporally persistent local high and low throughfall inputs.

On the one hand, some studies, mostly conducted in the arid regions and coniferous forests, reported that soil wetting patterns were not or only partly linked to throughfall variation, despite recurrent throughfall patterns (Raat et al., 2002; Shachnovich et al., 2008; Zhu et al., 2021). Forest floor thickness, horizontal water flow, and soil properties were suggested as reasons for the decoupled patterns. On the other hand, some modeling and field studies conducted in temperate deciduous forests found that throughfall patterns influenced soil moisture response rather than soil water storage variability (Coenders-Gerrits et al., 2013; Metzger et al., 2017; Fischer et al., 2023). In those studies, possible reasons were attributed to local

processes such as preferential flow due to soil water repellency, the soil pore structure, or elevated root water uptake. Our results support that it is not root water uptake but preferential flow paths that likely decouples the throughfall and soil water patterns. In fact, Fischer et al., (2023) using independent throughfall and soil water content sampling designs, demonstrated that the signature of throughfall patterns dissipated in the post-event soil water variation. However, they detected the stronger influence of throughfall patterns in the soil moisture response to rainfall in the 2015 and 2016 growing seasons. The temporal variation in soil water content in the 2019 growing season was similar to the seasonal decline in soil water content in 2015 (Metzger et al., 2017). Dry soil conditions can lead to rapid drainage due to reduced water holding capability (Jost et al., 2004; Blume et al., 2009; Wiekenkamp et al., 2016; Demand et al., 2019; Molina et al., 2019) regardless of throughfall amount and its variation. Therefore, our findings support that the localized throughfall input likely enhances preferential flow because of low soil retention (Fischer et al., 2023) rather than local root water uptake.

As a result, the fast flow processes likely dominate how water is stored and transported at our site, erasing the throughfall distribution signature in soil water and root water uptake patterns. Our results also support that the spatial variation of throughfall affects drainage and subsurface flow (Keim et al., 2006; Blume et al., 2009; Guswa and Spence, 2012), and root activity does not alter canopy-attributed heterogeneity in drainage pathways (Guswa, 2012).

The second reason (2) is related to water-limitation conditions. In central Europe, 2019 was the second consecutive extremely dry summer (Boergens et al., 2020), which damaged beech forests (Obladen et al., 2021). On average, however, the potential evapotranspiration demand was met at the study site despite the low soil water storage. The ratio of root water uptake to potential evapotranspiration was mostly above 65%, which is within the expected range even in the absence of shallow groundwater storage (Nie et al., 2021). Hence, local biotic and abiotic factors determined the spatial variation of root water uptake during growing season rather than throughfall patterns. However, the discrepancy between daily potential evapotranspiration and root water uptake only increased as the soil in the sampled layers dried out, possibly due to a potential shift in the water uptake depth (see below).

## 4. 2) Relative and average soil wetness shapes root water uptake patterns

We found that spatial variation in soil water storage strongly regulates local water uptake such that wetter locations enhance root water uptake. This finding is in line with expectations as transpiration rate relies on soil water availability and distribution (Couvreur et al., 2014; Klein et al., 2014; Hildebrandt et al., 2016). Here, our results provide further support that root water uptake is likely to reduce the spatial variability in soil water storage as has been previously suggested (Hopmans and Bristow, 2002; Ivanov et al., 2010; Neumann and Cardon, 2012).

For a given meteorological condition, root-water uptake at a particular location is a function of water transport resistance between root and soil in addition to the soil-water potential (Cardon and Letey, 1992; Shani and Dudley, 1996; Lhomme, 1998). Both characteristics depend on local soil texture and soil moisture, and the latter in turn is affected by the local rate of uptake. Although bulk density is attributed to porous space and eventually water retention (Zacharias and Wessolek, 2007; Looy et al., 2017), we surprisingly found that the bulk density of the monitored soil layer did not affect local water uptake. In contrast, the combination of higher field capacity and low soil water probably hindered the local water uptake due to lower soil water retention. Differences in local soil properties regulate matric potential at a certain soil wetness. Thus, our result indicates that wetter locations may not always correspond to the same degree of matric potential and ease root water uptake due to the local field capacity. However, our findings suggest that solely soil properties were less important than other tested variables despite their control on the spatial distribution of soil moisture (Vereecken et al., 2022) and water accessibility for transpiration (Vereecken et al., 2007; Cai et al., 2018).

In addition, the spatial mean of soil water affected root water uptake patterns, yet the effect depended on the basal area, i.e., the size of neighboring trees. We found that as the study site dries out, local water uptake increased in locations with smaller basal areas. Conversely, wetter site conditions facilitate greater water uptake at locations with higher basal areas, i.e., dense clusters of or large trees. We interpret this as a sign that larger trees are likely to shift their water uptake to deeper soil layers to meet transpiration demands, beyond the monitored soil depth (37 cm), as follows: Higher basal area likely increases transpiration demand and enhances water uptake as long as water is available. At the same time, locations with higher basal area exhaust the water storage faster but are able to shift uptake to deeper layers where

Hydrology and Earth System Sciences
EGU
Discussions

soil water content is not measured in our monitoring setup. Beech trees have extensive root systems at shallower depths similar to other temperate tree species, such as European ash and sycamore maple (Kreuzwieser and Gessler, 2010; Brinkmann et al., 2019). However, in response to declining soil water content in the topsoil, temperate tree species can tap water from the deeper soil layers (Brinkmann et al., 2019; Agee et al., 2021; Seeger and Weiler, 2021) despite their shallow root system (Leuschner, 2020). Recently, Agee et al. (2021) used a three-dimensional water uptake model based on observations in temperate mixed-deciduous forest to show that water uptake is shifted to the deeper soil layers as soil moisture depletes, which is consistent with the field observations. Also, Krämer and Hölscher (2010) observed in beech and mixed deciduous stands that roots can extract water at depths down to 70 cm soil depth. Similarly, to our site, theirs had a shallow soil layer underlain by weathered limestone.

### 4.3) Tree species richness regulates root water uptake patterns

In addition to the basal area, we included the number of species and number of tree individuals in the linear mixed effects analysis to explore further the biotic drivers of root water uptake patterns. While the number of trees was unimportant, the number of species and the basal area, showed a significant interaction effect on the local water uptake. The result indicates that an increase in species richness leads to greater root water uptake, depending on the size and/or density of the neighboring trees: Higher basal area, combined with more species, elevates water uptake. In other words, the interactions among neighboring tree species strongly determine root water uptake patterns, and at the same basal area more water can be taken up in a diverse compared to a less diverse neighborhood.

In temperate forests, transpiration has been observed to change with tree species richness at the stand level (Krämer and Hölscher, 2010; Gebauer et al., 2012; Kunert et al., 2012; Meißner et al., 2012; Forrester, 2014). Although some studies indicate a positive relationships between tree diversity and water uptake rate (Forrester et al., 2010; Krämer and Hölscher, 2010; Kunert et al., 2012), tree species diversity is not always positively related to water uptake. While Krämer and Hölscher (2010) observed a positive correlation between water uptake and species richness of the plots in the upper soil layers during soil drying in 2006 at the same study site, Meißner et al. (2012) found no relationship between tree diversity and root water uptake in 2009. They attributed this finding to wetter soil conditions. In contrast, Lübbe et

al. (2016) observed a weak effect of diversity on transpiration in wetter soil conditions but not in drier conditions compared to previous studies (e.g., Pretzsch et al., 2013; del Río et al., 2014). Shortage of water can inflate competition mechanisms for water among tree species (González de Andrés et al., 2018; Vitali et al., 2018; Magh et al., 2020). Our results can be used to show that competition between neighboring tree species increases water uptake capacity at more diverse spots (Wambsganss et al., 2021). In addition, different co-existing tree species can facilitate resource uptake or reduce competition, depending on the temporal and spatial availability of the sources, which is often defined as complementarity (Forrester and Bauhus, 2016). As reviewed and listed by Silvertown et al. (2015), several studies suggest that co-existing tree species reduce competition for subsurface water sources by adopting different vertical root water uptake strategies, referred to as hydrological niche partitioning. In addition, trees can transport water from moist to dry parts of the soil layers through their roots (Neumann and Cardon, 2012). The mechanism is called hydraulic redistribution or hydraulic lift, which can provide water availability to the shallow roots in drier layers (Burgess et al., 1998; Jonard et al., 2011; Hafner et al., 2017; Lee et al., 2018; Rodríguez-Robles et al., 2020; Hafner et al., 2021). Hafner et al. (2021) found in an experiment with six temperate tree species, including the European beech, that the neighboring tree species diversity may not be important for exploiting water uptake through hydraulic redistribution. Both hydraulic niche partitioning and redistribution have been observed vertically, whereas horizontal patterns are largely unexplored in the context of niche partitioning (Hildebrandt, 2020). Our results do not provide direct evidence for either hydraulic redistribution or horizontal niche partitioning. However, they indicate that horizontal root water uptake patterns are regulated by species richness.

# 5) Conclusion

We investigated the factors that influence the spatial patterns of root water uptake by considering heterogeneity in throughfall and soil water. To that end, we acquired a comprehensive data set based on throughfall measurements paired with soil moisture sensors in a mixed deciduous forest. Soil and neighboring tree characteristics were also included in the linear mixed effects model. We found that variation in root water uptake did not correspond to throughfall. Wetter soil locations, poorly associated

with higher throughfall, increased local root water uptake. In contrast, how average soil water conditions modified root water uptake depended on the neighborhood basal area. As the site dried out, large trees likely took up water in deeper layers to meet transpiration demands. Furthermore, an increase in species diversity promoted root water uptake, similarly depending on the size of neighboring trees, suggesting active complementarity mechanisms in the forest stand. In conclusion, our results suggest that soil water distribution and neighboring tree characteristics regulate root water uptake patterns more than soil properties and throughfall variation.

**Acknowledgments**

This study is part of the Collaborative Research Centre (CRC 1076 AquaDiva) of the Friedrich Schiller University Jena, funded by the Deutsche Forschungsgemeinschaft (DFG, German Research Foundation)—SFB 1076—Project Number 218627073. We thank to AquaDiva subproject D03 for weather station (Reckenbuel) data. Also, people who contributed to installation of soil moisture sensors in the research site: Ricardo Ontiveros-Enriques, Bernd Ruppe, Danny Schelhorn, Josef Weckmüller. We thank the Hainich CZE site manager Robert Lehmann and the Hainich National Park. We thank the bachelor and master students Carla Peter, Xiaoyu Zhao, Stephan Bock for their contribution to throughfall sampling.

**Data availability**

The dataset is currently being prepared for publication in an official repository. The DOI will be published with the data at the latest when the data are published.

**Author contributions**

GD and AH designed the throughfall measurement setup, AH and JCM designed soil moisture measurement. GD conducted the field sampling with assistance from JF and the students listed in the Acknowledgments. GD analyzed the data, developed the linear mixed effects model, and analyzed the results with AH and AG. GD prepared the first version of the manuscript, and all authors contributed to discussions and the final version of the manuscript.

**Competing interests**

Anke Hildebrandt is part of the editorial board of HESS. The peer-review process was guided by an independent editor, and the authors have also no other competing interests to declare.

# 6) References

Agee, E., He, L., Bisht, G., Couvreur, V., Shahbaz, P., Meunier, F., Gough, C. M., Matheny, A. M., Bohrer, G., and Ivanov, V.: Root lateral interactions drive water uptake patterns under water limitation, Advances in Water Resources, 151, 103896, https://doi.org/10.1016/j.advwatres.2021.103896, 2021.

Bachmair, S., Weiler, M., and Troch, P. A.: Intercomparing hillslope hydrological dynamics: Spatio-temporal variability and vegetation cover effects, Water Resources Research, 48, https://doi.org/10.1029/2011WR011196, 2012.

Baroni, G., Ortuani, B., Facchi, A., and Gandolfi, C.: The role of vegetation and soil properties on the spatio-temporal variability of the surface soil moisture in a maize-cropped field, Journal of Hydrology, 489, 148–159, https://doi.org/10.1016/j.jhydrol.2013.03.007, 2013.

Bartoń, K.: MuMIn: Multi-Model Inference, 2020.

Bates, D., Mächler, M., Bolker, B., and Walker, S.: Fitting Linear Mixed-Effects Models Using **lme4**, J. Stat. Soft., 67, https://doi.org/10.18637/jss.v067.i01, 2015.

Blume, T., Zehe, E., and Bronstert, A.: Use of soil moisture dynamics and patterns at different spatio-temporal scales for the investigation of subsurface flow processes, Hydrology and Earth System Sciences, 13, 1215–1233, https://doi.org/10.5194/hess-13-1215-2009, 2009.

Boergens, E., Güntner, A., Dobslaw, H., and Dahle, C.: Quantifying the Central European Droughts in 2018 and 2019 With GRACE Follow-On, Geophysical Research Letters, 47, e2020GL087285, https://doi.org/10.1029/2020GL087285, 2020.

Bogena, H. R., Herbst, M., Huisman, J. A., Rosenbaum, U., Weuthen, A., and Vereecken, H.: Potential of Wireless Sensor Networks for Measuring Soil Water Content Variability, Vadose Zone Journal, 9, 1002–1013, https://doi.org/10.2136/vzj2009.0173, 2010.

Borchers, H. W.: pracma: Practical Numerical Math Functions, 2021.

Bouten, W., Heimovaara, T. J., and Tiktak, A.: Spatial patterns of throughfall and soil water dynamics in a Douglas fir stand, Water Resources Research, 28, 3227–3233, https://doi.org/10.1029/92WR01764, 1992.

Brinkmann, N., Eugster, W., Buchmann, N., and Kahmen, A.: Species-specific differences in water uptake depth of mature temperate trees vary with water availability in the soil, Plant Biology, 21, 71–81, https://doi.org/10.1111/plb.12907, 2019.

Brum, M., Vadeboncoeur, M. A., Ivanov, V., Asbjornsen, H., Saleska, S., Alves, L. F., Penha, D., Dias, J. D., Aragão, L. E. O. C., Barros, F., Bittencourt, P., Pereira, L., and Oliveira, R. S.: Hydrological niche segregation defines forest structure and drought tolerance strategies in a seasonal Amazon forest, Journal of Ecology, 107, 318–333, https://doi.org/10.1111/1365-2745.13022, 2019.

Burgess, S. S. O., Adams, M. A., Turner, N. C., and Ong, C. K.: The redistribution of soil water by tree root systems, Oecologia, 115, 306–311, https://doi.org/10.1007/s004420050521, 1998.

Cai, G., Vanderborght, J., Langensiepen, M., Schnepf, A., Hüging, H., and Vereecken, H.: Root growth, water uptake, and sap flow of winter wheat in response to different soil water conditions, Hydrol. Earth Syst. Sci., 22, 2449–2470, https://doi.org/10.5194/hess-22-2449-2018, 2018.

Cardon, G. E. and Letey, J.: Plant Water Uptake Terms Evaluated for Soil Water and Solute Movement Models, Soil Science Society of America Journal, 56, 1876–1880, https://doi.org/10.2136/sssaj1992.03615995005600060038x, 1992.

Carlyle-Moses, Darryl. E., Lishman, Chad. E., and McKee, Adam. J.: A preliminary evaluation of throughfall sampling techniques in a mature coniferous forest, Journal of Forestry Research, 25, 407–413, https://doi.org/10.1007/s11676-014-0468-8, 2014.

Coenders-Gerrits, A. M. J., Hopp, L., Savenije, H. H. G., and Pfister, L.: The effect of spatial throughfall patterns on soil moisture patterns at the hillslope scale, Hydrol. Earth Syst. Sci., 17, 1749–1763, https://doi.org/10.5194/hess-17-1749-2013, 2013.

Cosh, M. H., Jackson, T. J., Moran, S., and Bindlish, R.: Temporal persistence and stability of surface soil moisture in a semi-arid watershed, Remote Sensing of Environment, 112, 304–313, https://doi.org/10.1016/j.rse.2007.07.001, 2008.

Couvreur, V., Vanderborght, J., Beff, L., and Javaux, M.: Horizontal soil water potential heterogeneity: simplifying approaches for crop water dynamics models, Hydrology and Earth System Sciences, 18, 1723–1743, https://doi.org/10.5194/hess-18-1723-2014, 2014.

Crockford, R. H. and Richardson, D. P.: Partitioning of rainfall into throughfall, stemflow and interception: effect of forest type, ground cover and climate, Hydrological Processes, 14, 2903–2920, 2000.

Demand, D., Blume, T., and Weiler, M.: Spatio-temporal relevance and controls of preferential flow at the landscape scale, Hydrol. Earth Syst. Sci., 23, 4869–4889, https://doi.org/10.5194/hess-23-4869-2019, 2019.

Demir, G., Michalzik, B., Filipzik, J., Metzger, J., and Hildebrandt, A.: Spatial variation of grassland canopy affects soil wetting patterns and preferential flow, https://doi.org/10.22541/au.164970545.54927607/v1, 2022.

Dunkerley, D.: Stemflow on the woody parts of plants: dependence on rainfall intensity and event profile from laboratory simulations, Hydrological Processes, 28, 5469–5482, https://doi.org/10.1002/hyp.10050, 2014.

Emerman, S. H. and Dawson, T. E.: Hydraulic Lift and Its Influence on the Water Content of the Rhizosphere: An Example from Sugar Maple, Acer saccharum, Oecologia, 108, 273–278, 1996.

Fan, J., Oestergaard, K. T., Guyot, A., Jensen, D. G., and Lockington, D. A.: Spatial variability of throughfall and stemflow in an exotic pine plantation of subtropical coastal Australia, Hydrological Processes, 29, 793–804, https://doi.org/10.1002/hyp.10193, 2015.

Fischer, C., Metzger, J. C., Demir, G., Wutzler, T., and Hildebrandt, A.: Throughfall spatial patterns translate into spatial patterns of soil moisture dynamics – empirical evidence, Ecohydrology/Instruments and observation techniques, https://doi.org/10.5194/hess-2022-418, 2023.

Forrester, D. I.: The spatial and temporal dynamics of species interactions in mixed-species forests: From pattern to process, Forest Ecology and Management, 312, 282–292, https://doi.org/10.1016/j.foreco.2013.10.003, 2014.

Forrester, D. I. and Bauhus, J.: A Review of Processes Behind Diversity—Productivity Relationships in Forests, Curr Forestry Rep, 2, 45–61, https://doi.org/10.1007/s40725-016-0031-2, 2016.

Forrester, D. I., Theiveyanathan, S., Collopy, J. J., and Marcar, N. E.: Enhanced water use efficiency in a mixed Eucalyptus globulus and Acacia mearnsii plantation, Forest Ecology and Management, 259, 1761–1770, https://doi.org/10.1016/j.foreco.2009.07.036, 2010.

Gebauer, T., Horna, V., and Leuschner, C.: Canopy transpiration of pure and mixed forest stands with variable abundance of European beech, Journal of Hydrology, 442–443, 2–14, https://doi.org/10.1016/j.jhydrol.2012.03.009, 2012.

Gerrits, A. M. J., Pfister, L., and Savenije, H. H. G.: Spatial and temporal variability of canopy and forest floor interception in a beech forest, Hydrol. Process., 24, 3011–3025, https://doi.org/10.1002/hyp.7712, 2010.

González de Andrés, E., Camarero, J. J., Blanco, J. A., Imbert, J. B., Lo, Y.-H., Sangüesa-Barreda, G., and Castillo, F. J.: Tree-to-tree competition in mixed European beech–Scots pine forests has different impacts on growth and water-use efficiency depending on site conditions, Journal of Ecology, 106, 59–75, https://doi.org/10.1111/1365-2745.12813, 2018.

Grayson, R. B., Western, A. W., Chiew, F. H. S., and Blöschl, G.: Preferred states in spatial soil moisture patterns: Local and nonlocal controls, Water Resources Research, 33, 2897–2908, https://doi.org/10.1029/97WR02174, 1997.

Guderle, M. and Hildebrandt, A.: Using measured soil water contents to estimate evapotranspiration and root water uptake profiles – a comparative study, Hydrol. Earth Syst. Sci., 17, 2015.

Guderle, M., Bachmann, D., Milcu, A., Gockele, A., Bechmann, M., Fischer, C., Roscher, C., Landais, D., Ravel, O., Devidal, S., Roy, J., Gessler, A., Buchmann, N., Weigelt, A., and Hildebrandt, A.: Dynamic niche partitioning in root water uptake facilitates efficient water use in more diverse grassland plant communities, Funct Ecol, 32, 214–227, https://doi.org/10.1111/1365-2435.12948, 2018.

Guo, J. S., Hungate, B. A., Kolb, T. E., and Koch, G. W.: Water source niche overlap increases with site moisture availability in woody perennials, Plant Ecol, 219, 719–735, https://doi.org/10.1007/s11258-018-0829-z, 2018.

Guswa, A. J.: Canopy vs. Roots: Production and Destruction of Variability in Soil Moisture and Hydrologic Fluxes, Vadose Zone Journal, 11, vzj2011.0159, https://doi.org/10.2136/vzj2011.0159, 2012.

Guswa, A. J. and Spence, C. M.: Effect of throughfall variability on recharge: application to hemlock and deciduous forests in western Massachusetts, Ecohydrology, 5, 563–574, https://doi.org/10.1002/eco.281, 2012.

Hafner, B. D., Tomasella, M., Häberle, K.-H., Goebel, M., Matyssek, R., and Grams, T. E. E.: Hydraulic redistribution under moderate drought among English oak, European beech and Norway spruce determined by deuterium isotope labeling in a split-root experiment, Tree Physiology, 37, 950–960, https://doi.org/10.1093/treephys/tpx050, 2017.

Hafner, B. D., Hesse, B. D., and Grams, T. E. E.: Friendly neighbours: Hydraulic redistribution accounts for one quarter of water used by neighbouring drought stressed tree saplings, Plant, Cell & Environment, 44, 1243–1256, https://doi.org/10.1111/pce.13852, 2021.

Hildebrandt, A.: Root-Water Relations and Interactions in Mixed Forest Settings, in: Forest-Water Interactions, edited by: Levia, D. F., Carlyle-Moses, D. E., Iida, S., Michalzik, B., Nanko, K., and Tischer, A., Springer International Publishing, Cham, 319–348, https://doi.org/10.1007/978-3-030-26086-6_14, 2020.

Hildebrandt, A., Kleidon, A., and Bechmann, M.: A thermodynamic formulation of root water uptake, Hydrol. Earth Syst. Sci., 14, 2016.

Hopmans, J. W. and Bristow, K. L.: Current Capabilities and Future Needs of Root Water and Nutrient Uptake Modeling, in: Advances in Agronomy, vol. 77, Elsevier, 103–183, https://doi.org/10.1016/S0065-2113(02)77014-4, 2002.

Hupet, F. and Vanclooster, M.: Micro-variability of hydrological processes at the maize row scale: implications for soil water content measurements and evapotranspiration estimates, Journal of Hydrology, 303, 247–270, https://doi.org/10.1016/j.jhydrol.2004.07.017, 2005.

Hupet, F., Lambot, S., Javaux, M., and Vanclooster, M.: On the identification of macroscopic root water uptake parameters from soil water content observations, Water Resources Research, 38, 36-1-36–14, https://doi.org/10.1029/2002WR001556, 2002.

IUSS Working Group, W. and others: World reference base for soil resources, World Soil Resources Report, 103, 2006.

Ivanov, V. Y., Fatichi, S., Jenerette, G. D., Espeleta, J. F., Troch, P. A., and Huxman, T. E.: Hysteresis of soil moisture spatial heterogeneity and the "homogenizing" effect of vegetation, Water Resources Research, 46, https://doi.org/10.1029/2009WR008611, 2010.

Jackisch, C., Knoblauch, S., Blume, T., Zehe, E., and Hassler, S. K.: Estimates of tree root water uptake from soil moisture profile dynamics, Biogeosciences, 17, 5787–5808, https://doi.org/10.5194/bg-17-5787-2020, 2020.

Jarecke, K. M., Bladon, K. D., and Wondzell, S. M.: The Influence of Local and Nonlocal Factors on Soil Water Content in a Steep Forested Catchment, Water Resources Research, 57, e2020WR028343, https://doi.org/10.1029/2020WR028343, 2021.

Jonard, F., André, F., Ponette, Q., Vincke, C., and Jonard, M.: Sap flux density and stomatal conductance of European beech and common oak trees in pure and mixed stands during the summer drought of 2003, Journal of Hydrology, 409, 371–381, https://doi.org/10.1016/j.jhydrol.2011.08.032, 2011.

Jost, G., Schume, H., and Hager, H.: Factors controlling soil water-recharge in a mixed European beech (Fagus sylvatica L.)–Norway spruce [Picea abies (L.) Karst.] stand, Eur J Forest Res, 123, 93–104, https://doi.org/10.1007/s10342-004-0033-7, 2004.

Katul, G. G. and Siqueira, M. B.: Biotic and abiotic factors act in coordination to amplify hydraulic redistribution and lift, The New Phytologist, 187, 3–6, 2010.

Keim, R. F., Skaugset, A. E., and Weiler, M.: Temporal persistence of spatial patterns in throughfall, Journal of Hydrology, 314, 263–274, https://doi.org/10.1016/j.jhydrol.2005.03.021, 2005.

Keim, R. F., Skaugset, A. E., and Weiler, M.: Storage of water on vegetation under simulated rainfall of varying intensity, Advances in Water Resources, 29, 974–986, https://doi.org/10.1016/j.advwatres.2005.07.017, 2006.

Kirchen, G., Calvaruso, C., Granier, A., Redon, P.-O., Van der Heijden, G., Bréda, N., and Turpault, M.-P.: Local soil type variability controls the water budget and stand productivity in a beech forest, Forest Ecology and Management, 390, 89–103, https://doi.org/10.1016/j.foreco.2016.12.024, 2017.

Kleidon, A. and Renner, M.: Thermodynamic limits of hydrologic cycling within the Earth system: concepts, estimates and implications, Hydrol. Earth Syst. Sci., 17, 2873–2892, https://doi.org/10.5194/hess-17-2873-2013, 2013.

Klein, T., Rotenberg, E., Cohen-Hilaleh, E., Raz-Yaseef, N., Tatarinov, F., Preisler, Y., Ogée, J., Cohen, S., and Yakir, D.: Quantifying transpirable soil water and its relations to tree water use dynamics in a water-limited pine forest, Ecohydrology, 7, 409–419, https://doi.org/10.1002/eco.1360, 2014.

Kohlhepp, B., Lehmann, R., Seeber, P., Küsel, K., Trumbore, S. E., and Totsche, K. U.: Aquifer configuration and geostructural links control the groundwater quality in thin-bedded carbonate–siliciclastic alternations of the Hainich CZE, central Germany, Hydrol. Earth Syst. Sci., 21, 6091–6116, https://doi.org/10.5194/hess-21-6091-2017, 2017.

Krämer, I. and Hölscher, D.: Soil water dynamics along a tree diversity gradient in a deciduous forest in Central Germany, Ecohydrology, 3, 262–271, https://doi.org/10.1002/eco.103, 2010.

Kreuzwieser, J. and Gessler, A.: Global climate change and tree nutrition: influence of water availability, Tree Physiology, 30, 1221–1234, https://doi.org/10.1093/treephys/tpq055, 2010.

Kühnhammer, K., Kübert, A., Brüggemann, N., Deseano Diaz, P., van Dusschoten, D., Javaux, M., Merz, S., Vereecken, H., Dubbert, M., and Rothfuss, Y.: Investigating the root plasticity response of Centaurea jacea to soil water availability changes from isotopic analysis, New Phytologist, 226, 98–110, https://doi.org/10.1111/nph.16352, 2020.

Kunert, N., Schwendenmann, L., Potvin, C., and Hölscher, D.: Tree diversity enhances tree transpiration in a Panamanian forest plantation, Journal of Applied Ecology, 49, 135–144, https://doi.org/10.1111/j.1365-2664.2011.02065.x, 2012.

Küsel, K., Totsche, K. U., Trumbore, S. E., Lehmann, R., Steinhäuser, C., and Herrmann, M.: How Deep Can Surface Signals Be Traced in the Critical Zone? Merging Biodiversity with Biogeochemistry Research in a Central German Muschelkalk Landscape, Frontiers in Earth Science, 4, https://doi.org/10.3389/feart.2016.00032, 2016.

Lee, E., Kumar, P., Barron-Gafford, G. A., Hendryx, S. M., Sanchez-Cañete, E. P., Minor, R. L., Colella, T., and Scott, R. L.: Impact of Hydraulic Redistribution on Multispecies Vegetation Water Use in a

Semiarid Savanna Ecosystem: An Experimental and Modeling Synthesis, Water Resour. Res., 54, 4009–4027, https://doi.org/10.1029/2017WR021006, 2018.

Leuschner, C.: Drought response of European beech (Fagus sylvatica L.)—A review, Perspectives in Plant Ecology, Evolution and Systematics, 47, 125576, https://doi.org/10.1016/j.ppees.2020.125576, 2020.

Levia, D. F. and Frost, E. E.: A review and evaluation of stemflow literature in the hydrologic and biogeochemical cycles of forested and agricultural ecosystems, Journal of Hydrology, 274, 1–29, https://doi.org/10.1016/S0022-1694(02)00399-2, 2003.

Levia, D. F. and Frost, E. E.: Variability of throughfall volume and solute inputs in wooded ecosystems, Progress in Physical Geography: Earth and Environment, 30, 605–632, https://doi.org/10.1177/0309133306071145, 2006.

Levia, D. F., Keim, R. F., Carlyle-Moses, D. E., and Frost, E. E.: Throughfall and Stemflow in Wooded Ecosystems, in: Forest Hydrology and Biogeochemistry: Synthesis of Past Research and Future Directions, edited by: Levia, D. F., Carlyle-Moses, D., and Tanaka, T., Springer Netherlands, Dordrecht, 425–443, https://doi.org/10.1007/978-94-007-1363-5_21, 2011.

Levia, D. F., Hudson, S. A., Llorens, P., and Nanko, K.: Throughfall drop size distributions: a review and prospectus for future research: Throughfall drop size distributions, WIREs Water, 4, e1225, https://doi.org/10.1002/wat2.1225, 2017.

Lhomme, J.-P.: Formulation of root water uptake in a multi-layer soil-plant model: does van den Honert's equation hold?, Hydrology and Earth System Sciences, 2, 31–39, https://doi.org/10.5194/hess-2-31-1998, 1998.

Looy, K. V., Bouma, J., Herbst, M., Koestel, J., Minasny, B., Mishra, U., Montzka, C., Nemes, A., Pachepsky, Y. A., Padarian, J., Schaap, M. G., Tóth, B., Verhoef, A., Vanderborght, J., Ploeg, M. J. van der, Weihermüller, L., Zacharias, S., Zhang, Y., and Vereecken, H.: Pedotransfer Functions in Earth System Science: Challenges and Perspectives, Reviews of Geophysics, 55, 1199–1256, https://doi.org/10.1002/2017RG000581, 2017.

Lübbe, T., Schuldt, B., Coners, H., and Leuschner, C.: Species diversity and identity effects on the water consumption of tree sapling assemblages under ample and limited water supply, Oikos, 125, 86–97, https://doi.org/10.1111/oik.02367, 2016.

Lüdecke, D., Ben-Shachar, M., Patil, I., Waggoner, P., and Makowski, D.: performance: An R Package for Assessment, Comparison and Testing of Statistical Models, JOSS, 6, 3139, https://doi.org/10.21105/joss.03139, 2021.

Magh, R.-K., Eiferle, C., Burzlaff, T., Dannenmann, M., Rennenberg, H., and Dubbert, M.: Competition for water rather than facilitation in mixed beech-fir forests after drying-wetting cycle, Journal of Hydrology, 587, 124944, https://doi.org/10.1016/j.jhydrol.2020.124944, 2020.

Magliano, P. N., Whitworth-Hulse, J. I., Florio, E. L., Aguirre, E. C., and Blanco, L. J.: Interception loss, throughfall and stemflow by Larrea divaricata: The role of rainfall characteristics and plant morphological attributes, Ecological Research, 34, 753–764, https://doi.org/10.1111/1440-1703.12036, 2019.

Martínez García, G., Pachepsky, Y. A., and Vereecken, H.: Effect of soil hydraulic properties on the relationship between the spatial mean and variability of soil moisture, Journal of Hydrology, 516, 154–160, https://doi.org/10.1016/j.jhydrol.2014.01.069, 2014.

Meißner, M., Köhler, M., Schwendenmann, L., and Hölscher, D.: Partitioning of soil water among canopy trees during a soil desiccation period in a temperate mixed forest, Biogeosciences, 9, 3465–3474, https://doi.org/10.5194/bg-9-3465-2012, 2012.

Metzger, J. C., Wutzler, T., Valle, N. D., Filipzik, J., Grauer, C., Lehmann, R., Roggenbuck, M., Schelhorn, D., Weckmüller, J., Küsel, K., Totsche, K. U., Trumbore, S., and Hildebrandt, A.: Vegetation impacts soil water content patterns by shaping canopy water fluxes and soil properties, Hydrological Processes, 31, 3783–3795, https://doi.org/10.1002/hyp.11274, 2017.

Metzger, J. C., Filipzik, J., Michalzik, B., and Hildebrandt, A.: Stemflow Infiltration Hotspots Create Soil Microsites Near Tree Stems in an Unmanaged Mixed Beech Forest, Front. For. Glob. Change, 4, 701293, https://doi.org/10.3389/ffgc.2021.701293, 2021.

Molina, A. J., Llorens, P., Garcia-Estringana, P., Moreno de las Heras, M., Cayuela, C., Gallart, F., and Latron, J.: Contributions of throughfall, forest and soil characteristics to near-surface soil water-content variability at the plot scale in a mountainous Mediterranean area, Science of The Total Environment, 647, 1421–1432, https://doi.org/10.1016/j.scitotenv.2018.08.020, 2019.

Nadezhdina, N., Cermak, J., Meiresonne, L., and Ceulemans, R.: Transpiration of Scots pine in Flanders growing on soil with irregular substratum, Forest Ecology and Management, 9, 2007.

Neumann, R. B. and Cardon, Z. G.: The magnitude of hydraulic redistribution by plant roots: a review and synthesis of empirical and modeling studies, New Phytologist, 194, 337–352, https://doi.org/10.1111/j.1469-8137.2012.04088.x, 2012.

Nie, C., Huang, Y., Zhang, S., Yang, Y., Zhou, S., Lin, C., and Wang, G.: Effects of soil water content on forest ecosystem water use efficiency through changes in transpiration/evapotranspiration ratio, Agricultural and Forest Meteorology, 308–309, 108605, https://doi.org/10.1016/j.agrformet.2021.108605, 2021.

Obladen, N., Dechering, P., Skiadaresis, G., Tegel, W., Keßler, J., Höllerl, S., Kaps, S., Hertel, M., Dulamsuren, C., Seifert, T., Hirsch, M., and Seim, A.: Tree mortality of European beech and Norway spruce induced by 2018-2019 hot droughts in central Germany, Agricultural and Forest Meteorology, 307, 108482, https://doi.org/10.1016/j.agrformet.2021.108482, 2021.

Otto, J., Berveiller, D., Bréon, F.-M., Delpierre, N., Geppert, G., Granier, A., Jans, W., Knohl, A., Kuusk, A., Longdoz, B., Moors, E., Mund, M., Pinty, B., Schelhaas, M.-J., and Luyssaert, S.: Forest summer albedo is sensitive to species and thinning: how should we account for this in Earth system models?, Biogeosciences, 11, 2411–2427, https://doi.org/10.5194/bg-11-2411-2014, 2014.

Pearson, R. K.: Data cleaning for dynamic modeling and control, in: 1999 European Control Conference (ECC), 1999 European Control Conference (ECC), 2584–2589, https://doi.org/10.23919/ECC.1999.7099714, 1999.

Pretzsch, H., Schütze, G., and Uhl, E.: Resistance of European tree species to drought stress in mixed versus pure forests: evidence of stress release by inter-specific facilitation, Plant Biology, 15, 483–495, https://doi.org/10.1111/j.1438-8677.2012.00670.x, 2013.

Priyadarshini, K. V. R., Prins, H. H. T., de Bie, S., Heitkönig, I. M. A., Woodborne, S., Gort, G., Kirkman, K., Ludwig, F., Dawson, T. E., and de Kroon, H.: Seasonality of hydraulic redistribution by trees to grasses and changes in their water-source use that change tree-grass interactions: HYDRAULIC REDISTRIBUTION BY TREES TO GRASSES AND CHANGES IN THEIR WATER SOURCES, Ecohydrol., 9, 218–228, https://doi.org/10.1002/eco.1624, 2016.

Pypker, T. G., Levia, D. F., Staelens, J., and Van Stan, J. T.: Canopy Structure in Relation to Hydrological and Biogeochemical Fluxes, in: Forest Hydrology and Biogeochemistry: Synthesis of Past Research and Future Directions, edited by: Levia, D. F., Carlyle-Moses, D., and Tanaka, T., Springer Netherlands, Dordrecht, 371–388, https://doi.org/10.1007/978-94-007-1363-5_18, 2011.

R Core Team: R: The R Project for Statistical Computing, R Foundation for Statistical Computing, Vienna, Austria, 2021.

Raat, K. J., Draaijers, G. P. J., Schaap, M. G., Tietema, A., and Verstraten, J. M.: Spatial variability of throughfall water and chemistry and forest floor water content in a Douglas fir forest stand, Hydrol. Earth Syst. Sci., 6, 363–374, https://doi.org/10.5194/hess-6-363-2002, 2002.

del Río, M., Schütze, G., and Pretzsch, H.: Temporal variation of competition and facilitation in mixed species forests in Central Europe, Plant Biology, 16, 166–176, https://doi.org/10.1111/plb.12029, 2014.

Rodrigues, A. F., Terra, M. C. N. S., Mantovani, V. A., Cordeiro, N. G., Ribeiro, J. P. C., Guo, L., Nehren, U., Mello, J. M., and Mello, C. R.: Throughfall spatial variability in a neotropical forest: Have we correctly accounted for time stability?, Journal of Hydrology, 608, 127632, https://doi.org/10.1016/j.jhydrol.2022.127632, 2022.

Rodríguez-Robles, U., Arredondo, J. T., Huber-Sannwald, E., Yépez, E. A., and Ramos-Leal, J. A.: Coupled plant traits adapted to wetting/drying cycles of substrates co-define niche multidimensionality, Plant, Cell & Environment, 43, 2394–2408, https://doi.org/10.1111/pce.13837, 2020.

Rosenbaum, U., Bogena, H. R., Herbst, M., Huisman, J. A., Peterson, T. J., Weuthen, A., Western, A. W., and Vereecken, H.: Seasonal and event dynamics of spatial soil moisture patterns at the small catchment scale: DYNAMICS OF CATCHMENT-SCALE SOIL MOISTURE PATTERNS, Water Resour. Res., 48, https://doi.org/10.1029/2011WR011518, 2012.

Sadeghi, S. M. M., Gordon, D. A., and Van Stan II, J. T.: A Global Synthesis of Throughfall and Stemflow Hydrometeorology, in: Precipitation Partitioning by Vegetation: A Global Synthesis, edited by: Van Stan, I., John T., Gutmann, E., and Friesen, J., Springer International Publishing, Cham, 49–70, https://doi.org/10.1007/978-3-030-29702-2_4, 2020.

Schume, H., Jost, G., and Hager, H.: Soil water depletion and recharge patterns in mixed and pure forest stands of European beech and Norway spruce, Journal of Hydrology, 289, 258–274, https://doi.org/10.1016/j.jhydrol.2003.11.036, 2004.

Schwärzel, K., Menzer, A., Clausnitzer, F., Spank, U., Häntzschel, J., Grünwald, T., Köstner, B., Bernhofer, C., and Feger, K.-H.: Soil water content measurements deliver reliable estimates of water fluxes: A comparative study in a beech and a spruce stand in the Tharandt forest (Saxony, Germany), Agricultural and Forest Meteorology, 149, 1994–2006, https://doi.org/10.1016/j.agrformet.2009.07.006, 2009.

Seeger, S. and Weiler, M.: Temporal dynamics of tree xylem water isotopes: in situ monitoring and modeling, Biogeosciences, 18, 4603–4627, https://doi.org/10.5194/bg-18-4603-2021, 2021.

Shachnovich, Y., Berliner, P. R., and Bar, P.: Rainfall interception and spatial distribution of throughfall in a pine forest planted in an arid zone, Journal of Hydrology, 349, 168–177, https://doi.org/10.1016/j.jhydrol.2007.10.051, 2008.

Shani, U. and Dudley, L. M.: Modeling water uptake by roots under water and salt stress: Soil-based and crop response root sink terms, Plant Roots: The Hidden Half, 635–641, 1996.

Silvertown, J., Araya, Y., and Gowing, D.: Hydrological niches in terrestrial plant communities: a review, Journal of Ecology, 103, 93–108, https://doi.org/10.1111/1365-2745.12332, 2015.

Spanner, G. C., Gimenez, B. O., Wright, C. L., Menezes, V. S., Newman, B. D., Collins, A. D., Jardine, K. J., Negrón-Juárez, R. I., Lima, A. J. N., Rodrigues, J. R., Chambers, J. Q., Higuchi, N., and Warren, J. M.: Dry Season Transpiration and Soil Water Dynamics in the Central Amazon, Frontiers in Plant Science, 13, 2022.

Staelens, J., De Schrijver, A., Verheyen, K., and Verhoest, N. E. C.: Spatial variability and temporal stability of throughfall water under a dominant beech (Fagus sylvatica L.) tree in relationship to canopy cover, Journal of Hydrology, 330, 651–662, https://doi.org/10.1016/j.jhydrol.2006.04.032, 2006.

Staelens, J., De Schrijver, A., Verheyen, K., and Verhoest, N. E. C.: Rainfall partitioning into throughfall, stemflow, and interception within a single beech (Fagus sylvatica L.) canopy: influence of foliation, rain event characteristics, and meteorology, Hydrological Processes, 22, 33–45, https://doi.org/10.1002/hyp.6610, 2008.

Teuling, A. J. and Troch, P. A.: Improved understanding of soil moisture variability dynamics, Geophysical Research Letters, 32, https://doi.org/10.1029/2004GL021935, 2005.

Thieurmel, B. and Elmarhraoui, A.: suncalc: Compute Sun Position, Sunlight Phases, Moon Position and Lunar Phase, 2022.

Vachaud, G., Passerat De Silans, A., Balabanis, P., and Vauclin, M.: Temporal Stability of Spatially Measured Soil Water Probability Density Function, Soil Science Society of America Journal, 49, 822–828, https://doi.org/10.2136/sssaj1985.03615995004900040006x, 1985.

Van Stan, J. T., Siegert, C. M., Levia, D. F., and Scheick, C. E.: Effects of wind-driven rainfall on stemflow generation between codominant tree species with differing crown characteristics, Agricultural and Forest Meteorology, 151, 1277–1286, https://doi.org/10.1016/j.agrformet.2011.05.008, 2011.

Van Stan, J. T., Hildebrandt, A., Friesen, J., Metzger, J. C., and Yankine, S. A.: Spatial Variability and Temporal Stability of Local Net Precipitation Patterns, in: Precipitation Partitioning by Vegetation: A Global Synthesis, edited by: Van Stan, I., John T., Gutmann, E., and Friesen, J., Springer International Publishing, Cham, 89–104, https://doi.org/10.1007/978-3-030-29702-2_6, 2020.

Vereecken, H., Kamai, T., Harter, T., Kasteel, R., Hopmans, J., and Vanderborght, J.: Explaining soil moisture variability as a function of mean soil moisture: A stochastic unsaturated flow perspective, Geophysical Research Letters, 34, https://doi.org/10.1029/2007GL031813, 2007.

Vereecken, H., Amelung, W., Bauke, S. L., Bogena, H., Brüggemann, N., Montzka, C., Vanderborght, J., Bechtold, M., Blöschl, G., Carminati, A., Javaux, M., Konings, A. G., Kusche, J., Neuweiler, I., Or, D., Steele-Dunne, S., Verhoef, A., Young, M., and Zhang, Y.: Soil hydrology in the Earth system, Nat Rev Earth Environ, 3, 573–587, https://doi.org/10.1038/s43017-022-00324-6, 2022.

Vitali, V., Forrester, D. I., and Bauhus, J.: Know Your Neighbours: Drought Response of Norway Spruce, Silver Fir and Douglas Fir in Mixed Forests Depends on Species Identity and Diversity of Tree Neighbourhoods, Ecosystems, 21, 1215–1229, https://doi.org/10.1007/s10021-017-0214-0, 2018.

Volkmann, T. H. M., Haberer, K., Gessler, A., and Weiler, M.: High-resolution isotope measurements resolve rapid ecohydrological dynamics at the soil–plant interface, New Phytologist, 210, 839–849, https://doi.org/10.1111/nph.13868, 2016.

Wambsganss, J., Beyer, F., Freschet, G. T., Scherer-Lorenzen, M., and Bauhus, J.: Tree species mixing reduces biomass but increases length of absorptive fine roots in European forests, J Ecol, 109, 2678–2691, https://doi.org/10.1111/1365-2745.13675, 2021.

Whelan, M. J. and Anderson, J. M.: Modelling spatial patterns of throughfaU and interception loss in a Norway spruce (Picea abies) plantation at the plot scale, Journal of Hydrology, 186, 335–354, 1996.

Wiekenkamp, I., Huisman, J. A., Bogena, H. R., Lin, H. S., and Vereecken, H.: Spatial and temporal occurrence of preferential flow in a forested headwater catchment, Journal of Hydrology, 534, 139–149, https://doi.org/10.1016/j.jhydrol.2015.12.050, 2016.

Wullaert, H., Pohlert, T., Boy, J., Valarezo, C., and Wilcke, W.: Spatial throughfall heterogeneity in a montane rain forest in Ecuador: Extent, temporal stability and drivers, Journal of Hydrology, 377, 71–79, https://doi.org/10.1016/j.jhydrol.2009.08.001, 2009.

Yu, K. and D'Odorico, P.: Hydraulic lift as a determinant of tree–grass coexistence on savannas, New Phytologist, 207, 1038–1051, https://doi.org/10.1111/nph.13431, 2015.

Zacharias, S. and Wessolek, G.: Excluding Organic Matter Content from Pedotransfer Predictors of Soil Water Retention, Soil Science Society of America Journal, 71, 43–50, https://doi.org/10.2136/sssaj2006.0098, 2007.

Zehe, E., Graeff, T., Morgner, M., Bauer, A., and Bronstert, A.: Plot and field scale soil moisture dynamics and subsurface wetness control on runoff generation in a headwater in the Ore Mountains, Hydrol. Earth Syst. Sci., 14, 873–889, https://doi.org/10.5194/hess-14-873-2010, 2010.

Zhang, Y., Wang, X., Hu, R., and Pan, Y.: Throughfall and its spatial variability beneath xerophytic shrub canopies within water-limited arid desert ecosystems, Journal of Hydrology, 539, 406–416, https://doi.org/10.1016/j.jhydrol.2016.05.051, 2016.

Zhu, X., He, Z., Du, J., Chen, L., Lin, P., and Tian, Q.: Spatial heterogeneity of throughfall and its contributions to the variability in near-surface soil water-content in semiarid mountains of China, Forest Ecology and Management, 488, 119008, https://doi.org/10.1016/j.foreco.2021.119008, 2021.

Zimmermann, A., Zimmermann, B., and Elsenbeer, H.: Rainfall redistribution in a tropical forest: Spatial and temporal patterns, Water Resour. Res., 45, https://doi.org/10.1029/2008WR007470, 2009.

Zuur, A. F., Ieno, E. N., Walker, N., Saveliev, A. A., and Smith, G. M.: Mixed effects models and extensions in ecology with R, Springer New York, New York, NY, https://doi.org/10.1007/978-0-387-87458-6, 2009.