# Peer review of "Root water uptake patterns are controlled by tree species"

_Hydrology and Earth System Sciences, 2023_

## Author Comment (AC1)

**Response Letter:**

**Reviewer(s)' Comments to Author:**
**Reviewer: 1**
**Comments to the Author**

MS#: hess-2023-91

Title: Root water uptake patterns are controlled by tree species interactions and soil water variability

It has been a pleasure reading through this contribution. This research underlines the lack of research on how throughfall patterns influence root water uptake patterns. The authors propose to close this knowledge gap by examining the role of throughfall patterns, soil water variability, soil properties, and biotic factors on root water uptake patterns using a statistical model.

We would like to thank the reviewer for finding our contribution valuable and for enjoying reading our manuscript.

I will organize my comments following the structure, per section, of the manuscript.

INTRODUCTION

The introduction provides a detailed background on below-canopy precipitation, specifically focusing on throughfall. It covers previous research about throughfall, its spatial distribution, and its impact on soil moisture patterns.

1.      Line 62: Consider rephrasing the phrase "Previously proposed explanations" to "Previous studies have suggested". This would be more direct.

Thanks, the sentence will be rephrased.

2.   Line 86: The term "water scarcity" is introduced without any context or explanation. A brief explanation or definition would enhance understanding. In fact, this is the only part of the manuscript where "water scarcity" is mentioned.

We agree that "water scarcity" was used vaguely. Here, we meant that the term refers to lack of soil water during drought, which is what the cited studies refer to as well. We will revise the sentence.

MATERIALS AND METHODS

The Materials and Methods section is well-written. Some areas that could be improved to make it clearer and easier to understand follow:

1. Line 174: Provide units for variables, such as λ (latent heat of vaporization).

Thanks for the reminder, the units and the constant variables will be provided.

2. Line 196-197: Elaborate why only 56 sensor locations provided high-quality data, and why only 34 of these provided data for root water uptake estimation. What qualifies as 'high-quality data'?

We agree that this section needs further explanation and will revise the sentences accordingly. Here, the high-quality data refers to the data that have passed the quality control (Section 2.3.1), which means that the soil water content data (6 min time intervals) are flagged and cleaned of artificial jumps and drops, or duplicate time stamps of different values, long discontinuities in the measurements, and lack of temporal variation in the time series despite rain events. During the throughfall sampling period among all sensors, only 56 sensors provided data that could be used after identification and removal of the errors within the data. The number of sensors decreased to 34 because only these sensors provided data within the dry periods when the root water uptake method can be applied to both soil layers at the same time interval.

3. Linear Mixed Effects Model: Explain the terminology used (like 'random effects', 'fixed effects') for readers unfamiliar with these statistical methods.

We will add new informative sentences to the section for explaining the terminology.

4. Linear Mixed Effects Model: Add a few sentences as justification or rationale for including each of the factor (fixed and random) and their interactions (for fixed) in the model. The reader may be able to identify the rationale by referring to the texts in Introduction. But it pays to be redundant in the Methods so that it is clear to the reader which factors were included, and more importantly 'why'.

Agreed, we will elaborate on the rationale for including interaction terms by describing that in a linear mixed effects model, the relationship between the dependent variable and one predictor, as it depends on the level of another predictor, can be represented by the interaction term. Thus, the fixed effects interaction terms represent the combined effect of the interacting predictors on the dependent variable.

DISCUSSION

This discussion section is generally well-written, but there are some areas for improvement to enhance clarity and readability:

1. Structure and Organization: The text is divided into several subtopics, which makes it easier to follow. However, it could benefit from a clearer outline or roadmap at the start of the discussion section that provides an overview of what will be discussed.

We will adopt the suggestion and add a general summary of the findings - a roadmap for discussion - at the beginning of the section.

2. Data Presentation: Some results are mentioned without an explicit description of how they were obtained. For example, in lines 477 to 479, the authors state they found that bulk density of the monitored soil layer did not affect local water uptake, but there is no explanation of how this conclusion was reached. Providing a more detailed explanation would enhance the credibility of the findings.

Thank you for pointing this out, and we agree that a more detailed explanation would be necessary to remind the reader of the finding before discussing it and providing final arguments. We will revise the discussion subsections that lack explicit description accordingly.

3. Clearer takeaways: The section could benefit from more direct conclusions or 'takeaway' after discussing each main point. For instance, after discussing the influence of tree species richness on root water uptake patterns (lines 504 to 533), a one-sentence conclusion summarizing the main takeaway could be beneficial.

Thank you for the suggestion. we agree and we will include a conclusion/takeaway statement in the sub-section. Moreover, we will adopt the same strategy in the other sub-sections if the take home message of the section is not clearly stated.

Hypotheses and Expectations: It might be useful to explicitly state the original hypotheses or expectations before explaining how the results confirm or contradict them. This can provide the reader with a clearer understanding of the study's purpose and significance. Perhaps, these hypotheses can be explicitly stated towards the end of Introduction. The authors can then 'revisit' these hypotheses in Discussion following their presentation of results.

We see the benefit of stating hypotheses to guide the reader; however, we believe that stating research questions and hypotheses at the same time may lead to repetition and would be a slightly different writing style compared to the current version of the manuscript. In the current version, we have structured the results and discussion sections of the manuscript according to the explicitly stated research questions. We have discussed our results in light of the expectations based on the previous studies (e.g., L416, L467, L477). In addition, the reviewer's previous comments will help to revise the manuscript to clarify the communication of the purpose and significance of the study. Yet as the reviewer sees a need to state hypotheses, we may add hypotheses at the end of the Introduction and revisit them in the general section of the Discussion.

5. Broader context of the literature. This study appeared broadly consistent with the finding of Knighton, Singh, Evaristo (2019, DOI: 10.1029/2019GL085937), which showed that monoculture catchments dense with trees reliant on shallow soil water exhibited reduced transpiration losses compared to deep-rooted and mixed-species forests. It is an important confirmation to make considering that this study is based purely on a statistical framework whilst that of Knighton et al. (2019) was based on the Budyko framework.

Thank you for drawing our attention to this catchment scale study. The authors studied 139 catchments. It enriches and highlights our findings, which emphasize the complex interplay between tree species diversity, complementary mechanisms, and water uptake patterns, and is consistent not only with the plot-scale studies listed above, but also with the larger-scale studies.

6.      Broader context of the literature. Demir et al. may find use in placing their finding within the larger context of the topic (root water uptake studies) that used other techniques, particularly stable H and O isotopes in water. That is, the hours-long timescales used in this study for estimating transpiration losses are orders of magnitude shorter than what stable isotopes would show. See, for example, the study by Evaristo et al. (2019, DOI: 10.1029/2018WR023265), which showed that transpiration water was between 17 and 62 days. How do the timescales of Demir et al. (and their findings) compare and contrast to the timescales (and findings) of studies using tracers? A few sentences that place Demir et al. within the larger context of the topic would be useful for future researchers to recognize.

Thank you for the suggestion, which will definitely help to put our findings in a broader context, we will take it into account when revising the manuscript. With our study, we cannot estimate the exact travel time (or time scale) for water transport to the deeper layer to the ground water and water transpiration through trees: However, our findings are in line with Evaristo et al.(2019) and ecohydrological separation phenomenon such that our finding support that throughfall inputs are rapidly transported into deep layers with preferential flow paths while transpiration is mainly driven by water remained in the soil which in result pose to longer residency time. In other words, our study suggests that the main source for transpiration is the water remaining in the soil  with a longer residence time which concurs with the previous studies suggest that water is taken up by trees from soil matrix storage while ground water recharge is fed by active water storage – preferential flow or due to local soil structure (e.g, Evaristo et al., 2019; Sprenger et al., 2019).  We will clearly incorporate this statement in the subsection 4.2

---

## Author Comment (AC2)

**Response Letter:**

**Reviewer(s)' Comments to Author:**
**Comments to the Author**
MS#: hess-2023-91

Title: Root water uptake patterns are controlled by tree species interactions and soil water variability

The manuscript provides valuable insights for quantifying the role of throughfall patterns on root water uptake patterns and the drive factor of root water uptake patterns. It is of great significance to understand the effects of throughfall patterns on subsurface water dynamics and the interaction between plant-soil-water systems in forest ecosystems. The study seems interesting Overall, I would recommend this manuscript for publication in HESS after some redivision. The comments and suggestions as follow:

Thank you very much for reading the manuscript carefully and providing your feedback to improve it. We have tried to understand the essence of each comment and make propositions how to accommodate them in the revision.

**In the introduction section**,

The research methods of root water uptake patterns should be introduced so that we can better understand the subsequent analysis.

We understand, also from later comments, that the methodology on the root water uptake should at least be very briefly mentioned in the introduction. We will do so shortly, mentioning the water balance method and also comparing it to the other methods such as stable water isotope signatures.

Further statements on the effect of throughfall patterns on the variation of soil moisture should be added. For example, some previous studies showed that the weak and short-term influence of throughfall patterns on the soil moisture patterns. However, when the proportion of throughfall is greater, whether this relationship will change.

We are not entirely sure, whether we understand what the reviewer means. We believe the reviewer refers to a potential contradiction between previous results and the research question here, which we ask whether throughfall patterns affect root water uptake. The latter implies that throughfall patterns are retained in the soil in the first place, but previous research shows that this is rarely the case (Metzger et al., 2017). Yet this is an important point. Our research question was motivated by previous research on other sites (Zhu et al., 2021; Metzger et al., 2017; Coenders-Gerrits et al., 2013), and we wanted to address it directly. In addition, we see in a recent paper, (Fischer et al., 2023), that the throughfall patterns leave a slight imprint on the soil water content even after drainage, although other drivers are much stronger. We will add some sentences to the introduction to give more background on this point.

The influence of abiotic factors on water use patterns should also be mentioned.

Thank you for reminding us. We will add a few sentences to mention the effect of abiotic factors on root water uptake which will also improve the communication of the results

regarding abiotic factors that were included in the linear mixed effects model, such as soil bulk density, field capacity etc.

**In the Materials and methods section,**

What you mean" We measured gross precipitation and throughfall on rainless days"? It is confusing how you can measure total rainfall when there is no rain.

Yes, we understand now that the expression is confusing. What we meant to say is that we read out the collectors during a precipitation-free period. For the weekly sampling we sometimes shifted the days, to catch a rain-free period. We will revise the sentence as follows.

"Gross precipitation and throughfall was read out on days without rain. Thus, the sampling interval ranged between six and eight days depending on the occurrence of rain events."

**In 2.4 section, what indicators are used to reflect root water uptake?**

We are not sure, what the reviewer means. We derived root water uptake directly from the high-resolution measurements of volumetric soil water content (section 2.4). We checked whether the decrease of soil water storage during daylight hours was larger than that during the night. If this was the case, this decrease was taken as root water uptake. The method was introduced by (Guderle and Hildebrandt, 2015) and shown to perform well on high resolution weighted lysimeters (Guderle et al., 2018).

We will revise the first sentence of the section to improve communication:

"We estimated root water uptake using the multi-step, multi-layer regression method (MSML), which is a water balance method and derives evapotranspiration from diurnal differences in soil water content (Guderle and Hildebrandt, 2015; Guderle et al., 2018)."

**How to integrate the water uptake pattern of a single site to the plot scale?**

We are not sure we understand what the reviewer means. Probably there is a misunderstanding regarding the scale at which the root water uptake was calculated. Here we do not calculate the root water uptake on the plot or subplot scale but at the microsite scale. We have not attempted any upscaling. In this paper, we are interested in the differences between microsites, i.e., spatial patterns, which are characterized by the soil moisture sensor locations. We then analyze whether microsite root water uptake is higher or lower compared to the other sites using the spatial deviation from the mean (see Line 219-223, at section 2.3.3) We consider including a short explanation in the model section (2.5) that root water uptake was characterized by the sensor location so as not to mislead the reader about the spatial scale.

In the Discussion section:

L454-455: What exactly does root activity mean? How does this study prove that root activity does not alter canopy-attributed heterogeneity in drainage pathways?

We agree that the sentence is vaguely formulated and lacks a clear explanation leading to the question marks as the reviewer pointed out. We will revise the sentences by describing and including exclusive description of the root activity to address these questions.

In the cited study (Guswa, 2012) in the corresponding lines, root activity refers to transpiration, root compensation, and hydraulic redistribution. Guswa (2012), employed numerical experiments to investigate the competing effects of canopy and root processes at the patch scale on water balance, recharge localization, spatial and temporal variability of soil moisture, and the upscaled relationship between mean soil moisture and transpiration. In this study, the water balance and upscaled fluxes were shown to be relatively insensitive to horizontal variability.

How do you prove that bulk density of the monitored soil layer did not affect local water uptake?

This study focused on the effect of soil bulk density on root water uptake, but other soil properties, such as texture and organic matter, affect soil structure and aggregate characteristics, which will indirectly affect water transport in soil and thus affect the water use pattern of plants.

This comment touches on two points (1) how do we see that soil bulk density had no effect (2) even is soil bulk density was not relevant, other soil properties (not measured) may have been.

Regarding (1) we have a clear answer: Soil bulk density did not come out as a significant variable during model selection, and we therefore conclude that it was not an important driver in our setting.

Regarding (2) unfortunately, we do not have measurements of soil organic matter and soil texture at all locations (which really would have been a valuable but equally time-intensive dataset). Therefore, statistically speaking both may have influenced the uptake without us noticing and may be hidden in the random factors. However, from the soil physics perspective soil bulk density is dependent strongly both on texture and soil organic carbon and integrates these two soil properties, because they affect aggregation and soil structure. Therefore, we think that an effect of either texture or soil organic matter would have resulted in soil bulk density being significant, unless they compensate each other, which is unexpected. Note however, that the field capacity was significant and arguably influenced by aggregation as well. Therefore, some signal of soil properties may be contained there. We will add few sentences to the discussion.

Trees with different ages have different physiological structures, such as root system and leaf characteristics, which will affect the water use pattern of plants. The effects of different tree ages on the water use pattern of plants should be discussed.

Agreed. We can add a short section on how our results may have been driven by the demography of the surrounding stand.

References

Coenders-Gerrits, A. M. J., Hopp, L., Savenije, H. H. G., and Pfister, L.: The effect of spatial throughfall patterns on soil moisture patterns at the hillslope scale, Hydrol. Earth Syst. Sci., 17, 1749–1763, https://doi.org/10.5194/hess-17-1749-2013, 2013.

Fischer, C., Metzger, J. C., Demir, G., Wutzler, T., and Hildebrandt, A.: Throughfall spatial patterns translate into spatial patterns of soil moisture dynamics – empirical evidence, Ecohydrology/Instruments and observation techniques, https://doi.org/10.5194/hess-2022-418, 2023.

Guderle, M. and Hildebrandt, A.: Using measured soil water contents to estimate evapotranspiration and root water uptake profiles – a comparative study, Hydrol. Earth Syst. Sci., 17, 2015.

Guderle, M., Bachmann, D., Milcu, A., Gockele, A., Bechmann, M., Fischer, C., Roscher, C., Landais, D., Ravel, O., Devidal, S., Roy, J., Gessler, A., Buchmann, N., Weigelt, A., and Hildebrandt, A.: Dynamic niche partitioning in root water uptake facilitates efficient water use in more diverse grassland plant communities, Funct Ecol, 32, 214–227, https://doi.org/10.1111/1365-2435.12948, 2018.

Guswa, A. J.: Canopy vs. Roots: Production and Destruction of Variability in Soil Moisture and Hydrologic Fluxes, Vadose Zone Journal, 11, vzj2011.0159, https://doi.org/10.2136/vzj2011.0159, 2012.

Metzger, J. C., Wutzler, T., Valle, N. D., Filipzik, J., Grauer, C., Lehmann, R., Roggenbuck, M., Schelhorn, D., Weckmüller, J., Küsel, K., Totsche, K. U., Trumbore, S., and Hildebrandt, A.: Vegetation impacts soil water content patterns by shaping canopy water fluxes and soil properties, Hydrological Processes, 31, 3783–3795, https://doi.org/10.1002/hyp.11274, 2017.

Zhu, X., He, Z., Du, J., Chen, L., Lin, P., and Tian, Q.: Spatial heterogeneity of throughfall and its contributions to the variability in near-surface soil water-content in semiarid mountains of China, Forest Ecology and Management, 488, 119008, https://doi.org/10.1016/j.foreco.2021.119008, 2021.

---

## Author Response (AR1)

**Response Letter:**

Dear Editor and Reviewers,

Once again, we would like to thank you for the fruitful discussion to improve our manuscript. We have carefully considered all the comments and made the necessary changes to the document. We have also changed the citation of the preprint publication by Fischer-Bedtke et al. (2023), as it was accepted during the discussions-review processes. In addition to the reviewer recommendations, we worked on the readability of the paper, which resulted in some small changes in the sentences. These changes can be seen in the Author's track-changes file in a different color.

Please see below for a detailed explanation of the changes related to the reviewer comments. The line numbers are written according to the Author's track-changes file.

Best regards,

On behalf of the authors,

Gökben Demir

INTRODUCTION:

We have rewritten the sentences in line 64 (in the tracked pdf file) and revised the sentence in line 122 to eliminate the use of "water scarcity" as suggested by Reviewer 1. We also added information about the research methods for root water uptake patterns in the line between 101 and 113, as suggested by Reviewer 2. We added additional sentences and revised the paragraph to clarify the previous finding about the effect of throughfall patterns on soil moisture variation in the line between 64-83. We revised lines 84-95 to add sentences about abiotic factors for root water uptake patterns.

We also stated the main hypothesis of the study and revised it in the discussion section as suggested by the reviewer 1.

MATERIALS AND METHODS.

As recommended by the reviewer 1:

We not only worked on the readability of the section, but also provided the units for the variables in line 220-221, elaborated further on the data quality criteria and the reasons for discarding some of the sensors' data, please see lines between 241-246. To familiarize readers who may not have experience with the linear mixed effects model we added few sentences in the section explaining the terminology and the rationale for including the fixed effects and their interaction in the statistical model, we have revised lines between 309-317, 326-338.

In line with the suggestions of reviewer 2

We have revised the sentences that created confusion about manual throughfall sampling, see lines between 207- 211. We also revised subsection 2.4, which explains the calculation of root water uptake, to clarify the communication about how we determined the water uptake, and added a brief explanation in the model section (2.5) that the root water uptake was characterized by the sensor location, so as not to mislead the reader about the spatial scale.

DISCUSSION:

As we stated in response to reviewer 1 in the first letter, we have taken the following steps in the discussion section:

A summary - roadmap section at the beginning of the discussion, which is between lines 487 - 494.

We have included more detailed explanations of the results to remind the reader in the subsections, as an example see lines 584-598.

We have incorporated clearer takeaway suggestion by adding clearer and more concise statements about the results, such as in subsection 4.3 lines between 675-682.

We have clearly stated the expectations and revisited the main hypothesis and included broader context of the literature as in lines between 571-575, 675-682, 689-69.

As recommended by reviewer 2:

We clarified about the root activity, which was vaguely worded and lacked a clear explanation, by describing and including exclusive description of the root activity to address these questions. In the cited study (Guswa, 2012), in the corresponding lines, root activity refers to transpiration, root compensation, and hydraulic redistribution.

We further elaborated on how we evaluated the effect of bulk density on root water uptake, since it did not come out as a significant variable during model selection, and we therefore conclude that it was not an important driver in our setting. We revised this in subsection 4.2 and in the lines between 580-600.

Furthermore, we discussed the effects of other factors mentioned by the reviewer, such as tree age and different physiological structures in the subsection lines between 622-636.

---

## Author Response (AR2)

**Response Letter:**

Dear Editor and Reviewer,

Thank you for your help in improving the manuscript.

Since reviewer #1 accepted the manuscript as is, we have only addressed reviewer #2's comments and made necessary minor changes. We also thank Reviewer #2 for bringing the Introduction to our attention. We used the second round of the review process to improve and revise the Introduction and the Abstract beyond what the reviewer requested. In the revised version, the introduction is improved in terms of guiding the reader about the research question and emphasizing that the research objective is about root water uptake patterns rather than investigating throughfall patterns.

In addition, we address all of the reviewer's recommendations and comments below.

Sincerely, On behalf of the authors, Gökben Demir **Reviewer(s)' Comments to Author:**

Comments to Reviewer Report #2: MS#: hess-2023-91

" Root water uptake patterns are controlled by tree species interactions and soil water variability" by Demir et al.

Throughfall is the largest source of water entering the soil in forests, and its spatial distribution depends on several biotic and abiotic factors. This study explored the influencing factors that affect root water uptake, including throughfall rainfall, soil water, and tree species abundance. It seems reasonable and meaningful, but there are minor problems that need to be modified. The specific issues are as follows:

Thank you for reading the revised manuscript and providing feedback to improve it. We have tried to understand the essence of each comment, thoroughly addressed each comment, and made the necessary changes to the document. In the following, we respond to all questions and recommendations, which are written in blue.

Introduction:

There are too many descriptions of factors affecting root water uptake, please simplify it.

We are not sure what the reviewer means. However, we understand that the introduction could be improved and simplified for clearer communication. Therefore, we have taken the comment into account to revise the introduction. We worked on it to better guide the reader to the research questions and made necessary changes, which are highlighted in the following comments. For example, to simplify it, we shortened it by removing explanations on the effect of tree species on water uptake, as this is well covered in the discussion section. In addition, we have revised the abstract to improve clear communication of the take-home message of the paper. Nevertheless, we would like to draw attention to the previous reviewer reports that requested more details on the influence of abiotic factors on water uptake patterns. Thus, we have incorporated the previous reviewer requests for more information on abiotic factors affecting water uptake, such as soil properties, and kept them in the revised version.

Lines 62-75: This paragraph focuses on the impact of throughfall rainfall on soil moisture dynamics, and then points out that there has been no research on the effect of throughfall rainfall on root water uptake. However, the last sentence confuses me. "Therefore, it is unclear how water uptake patterns play a role in translating throughfall patterns into spatio-temporal variation of soil water and vice versa". This seems to mean that root water uptake affects the impact of throughfall on soil water, which is not consistent with the title.

Thank you for your attention, we see that this part may confuse the reader, so we have revised the paragraph to improve communication, please see lines between 52-80 in the revised version that provides the current state of knowledge on what are the proposed reasons for the short and weak impact of spatial variability of throughfall on soil moisture patterns.

Although throughfall patterns have the potential to alter soil moisture patterns both in the wetting phase and in the drained state, the field studies observed that the impact of throughfall distribution on soil moisture patterns ceases rapidly and weakens in the drained state. However, the potential effect of throughfall patterns on root water uptake is lacking in experimental studies.

Furthermore, we think that this paragraph does not conflict with the title of the manuscript because the title mentions the main drivers of root water uptake patterns, as our results showed that throughfall variability does not translate into root water uptake patterns and soil water patterns.

Lines 84-85: I think caution needs to be expressed here. Variations in soil moisture do not reflect water uptake by the roots. There may be other reasons.

We agree. This is what we refer to with the surrounding text. In line with the comment, we revised to improve the text in an inclusive way for all possible reasons, please see Lines 71-95 in the revised version. Moreover, we restructured this part of the introduction to improve introduction together with the former comments.

**Table 2: Why are the dates out of order?**

As stated in the table caption, the throughfall data is ordered by the size of the gross precipitation. As the spatial variation of throughfall changes according to the size of the event (Levia and Frost, 2006; Levia et al., 2019; Metzger et al., 2017; Keim and Link, 2018). In this study, the spatial variation of throughfall is more important than the chronological order of the field data. In this study, we investigated how the variation in water input affects the variation in root water uptake. Therefore, the data is listed according to the gross precipitation amount, and we decide to keep it as is.

Figure 3: Why are the abscissas in Figure 3 inconsistent?

Thank you for the comment, we revised the caption to avoid further confusion for future readers. Figure 3 shows the spatial deviation of different variables (throughfall, soil water, and root water uptake). Since the spatial patterns of these values are different, the ordered values are different at each sensor location. However, the coloring scheme (light to dark) is kept the same based on throughfall (water input) as written in the figure caption. The darker colored throughfall locations are also dark in the other plots, showing that more water input than the spatial average does not necessarily correspond to more water uptake or more water stored in the soil. If we saw the same patterns across all variables in this plot, the x-axis would be identical. However, the plot shows that neither soil water and throughfall, throughfall and root water uptake, nor variations in soil water and root water uptake are directly related, which was also communicated in lines 390 - 398.

**Result:**

**What is the relationship between section 3.2 and 3.3? Where is section 3.4?**

Thanks for the point out, there is a typo in the section header, so section 3.5 should be 3.4, we have changed the number in the section title.

We are unsure what reviewer requested or questioned regarding the relationship between section 3.2 and 3.3.

In the first round of discussion, reviewers (see previous reviewers' reports) found the separate sections in the results part nice to read and stated that the structure helped them to follow the results. Furthermore, in the first round, we provided a short summary of the main results at the beginning of the discussion, as requested by the reviewers. Here, if the reviewer request on a short, small introductory paragraph at the beginning of the results, we think that it could make the text repetitive because of the small introductory paragraph in the discussion section. So we decided to leave it as it is for now. If the editor and reviewer think that we should combine these two sections, we can make this small change during the proofreading stage, but we do not think that it would greatly improve the text. Because here:

Section 3.2 gives the spatial average of soil water storage, potential evapotranspiration, and root water uptake along with the water holding capacity, which is how much water is available to root systems and what the atmospheric demand is, so it gives a background picture of the temporal changes in the values along with the quartile-based variation of the values.

Section 3.3, on the other hand, presents the results of the spatial deviation from the mean, which is also used in the literature to estimate the temporal stability of patterns, and is different from the quartile coefficient of variation.

Figure 4 and 5: What is the difference between the two figures, and is it necessary if it is just for visualization?

While Figure 4 shows the significant drivers, Figure 5 explains the interaction terms, which provide detailed information about how one fixed effect changes the impact of the other fixed effect on the estimated variable, so Figure 5 adds information to Figure 4. The plot type changes because interactions are best interpreted visually. Therefore, we decided to keep Figure 5 in the main manuscript.

Lines 488-490: Regarding the impact of throughfall on root water uptake, the manuscript mentioned that the lower soil contains more clay particles in the Material and Methods and should be difficult to drainage, and the discussion also mentioned that local rainfall input increases the preferential flow path. Is this contradictory? Will the roots utilize deeper soil water?

Relatively higher clay content in the subsoil does not prevent the occurrence of preferential flow (e.g., Jarvis et al., 2016; Nimmo, 2021), so it is not inconsistent that despite the higher clay content, the locally increased water input facilitates preferential flow particularly after dry spells. The linear mixed effects model suggests that trees may shift water uptake to deeper layers, depending on the average wetness of the site, as discussed in section.

**References:**

Jarvis, N., Koestel, J., and Larsbo, M.: Understanding Preferential Flow in the Vadose Zone: Recent Advances and Future Prospects, Vadose Zone Journal, 15, vzj2016.09.0075, https://doi.org/10.2136/vzj2016.09.0075, 2016.

Keim, R. F. and Link, T. E.: Linked spatial variability of throughfall amount and intensity during rainfall in a coniferous forest, Agricultural and Forest Meteorology, 248, 15–21, https://doi.org/10.1016/j.agrformet.2017.09.006, 2018.

Levia, D. F. and Frost, E. E.: Variability of throughfall volume and solute inputs in wooded ecosystems, Progress in Physical Geography: Earth and Environment, 30, 605–632, https://doi.org/10.1177/0309133306071145, 2006.

Levia, D. F., Nanko, K., Amasaki, H., Giambelluca, T. W., Hotta, N., Iida, S., Mudd, R. G., Nullet, M. A., Sakai, N., Shinohara, Y., Sun, X., Suzuki, M., Tanaka, N., Tantasirin, C., and Yamada, K.: Throughfall partitioning by trees, Hydrological Processes, 33, 1698–1708, https://doi.org/10.1002/hyp.13432, 2019.

Metzger, J. C., Wutzler, T., Valle, N. D., Filipzik, J., Grauer, C., Lehmann, R., Roggenbuck, M., Schelhorn, D., Weckmüller, J., Küsel, K., Totsche, K. U., Trumbore, S., and Hildebrandt, A.: Vegetation impacts soil water content patterns by shaping canopy water fluxes and soil properties, Hydrological Processes, 31, 3783–3795, https://doi.org/10.1002/hyp.11274, 2017.

Nimmo, J. R.: The processes of preferential flow in the unsaturated zone, Soil Science Society of America Journal, 85, 1–27, https://doi.org/10.1002/saj2.20143, 2021.